# Likelihood Annealing:
# Fast Calibrated Uncertainty for Regression

## Abstract

Recent advances in deep learning have shown that uncertainty estimation is becoming increasingly important in applications such as medical imaging, natural language processing, and autonomous systems. However, accurately quantifying uncertainty remains a challenging problem, especially in regression tasks where the output space is continuous. Deep learning approaches that allow uncertainty estimation for regression problems often converge slowly and yield poorly calibrated uncertainty estimates that can not be effectively used for quantification. Recently proposed post hoc calibration techniques are seldom applicable to regression problems and often add overhead to an already slow model training phase. This work presents a fast calibrated uncertainty estimation method for regression tasks called *Likelihood Annealing*, that consistently improves the convergence of deep regression models and yields calibrated uncertainty without any post hoc calibration phase. Unlike previous methods for calibrated uncertainty in regression that focus only on low-dimensional regression problems, our method works well on a broad spectrum of regression problems, including high-dimensional regression. Our empirical analysis shows that our approach is generalizable to various network architectures, including multilayer perceptrons, 1D/2D convolutional networks, and graph neural networks, on five vastly diverse tasks, i.e., chaotic particle trajectory denoising, physical property prediction of molecules using 3D atomistic representation, natural image super-resolution, and medical image translation using MRI.

## 1   Introduction

Uncertainty estimation is an essential building block to provide interpretability and secure reliability in modern machine learning systems (Shafaei et al., 2018; Kläs & Vollmer, 2018; Varshney & Alemzadeh, 2017; Hüllermeier & Waegeman, 2021) that offer intelligent solutions for numerous real-world applications, ranging from medical analytics (Leibig et al., 2017; Gillmann et al., 2021; Upadhyay et al., 2021b) to autonomous driving (Xu et al., 2014; Shafaei et al., 2018; Besnier et al., 2021). Recent advances have explored various formulations to provide accurate predictions and uncertainty estimates for deep neural networks, as represented by Bayesian approaches (Gal & Ghahramani, 2016; Kendall & Gal, 2017; Maddox et al., 2019), ensembles (Lakshminarayanan et al., 2017), pseudo-ensembles (Mehrtash et al., 2020; Franchi et al., 2020), and quantile regression (Romano et al., 2019; Yan et al., 2018; Feldman et al., 2021) methods. However, these existing methods are often computationally expensive – e.g., slow convergence rate during training or inefficient inference cost due to multiple forward passes – while being poorly calibrated for uncertainty estimates. Moreover, some of these methods are proposed for low-dimensional regression tasks (Chung et al., 2021; Zhou et al., 2021; Chen et al., 2021) (i.e., regressing a scalar value) and do not scale for high-dimensional regression (i.e., regressing large matrices or tensors). This paper presents a unified formulation to resolve these issues for estimating fast, well-calibrated uncertainty in deep regression models for a wide spectrum of regression problems, including chaotic particle trajectory denoising, physical property prediction of molecules using 3D atomistic representation, natural image super-resolution, and medical image translation using MRI.

We propose to revisit deep regression models trained via maximum likelihood estimation (MLE), which assumes a Gaussian distribution over the regression output and optimizes the negative log-likelihood to estimate the target and uncertainty. Although such models can ensure low regression error (i.e., high accuracy)

and encapsulate the predictive uncertainty, they often converge slowly at the beginning of training due to a flat gradient landscape. Further, they may even risk gradient explosion caused by a steep gradient landscape when reaching the optima (detailed in Section 3.1), leading to poorly calibrated uncertainty estimates that do not offer credible interpretability for the model and cannot be used for downstream applications.

To reshape the aforementioned ill-posed gradient landscape that causes slow convergence and poorly calibrated uncertainty, we propose a novel *Likelihood Annealing* (LIKA) scheme for deep regression models that alters the original gradients by formulating a temperature-dependent improper likelihood to be optimized during the learning phase. In contrast to the standard likelihood for regression that enforces a fixed Gaussian distribution on the target, we introduce a temperature hyperparameter to impose an evolving distribution.

The proposed temperature-dependent likelihood brings crucial properties to regression uncertainty. First, the multimodal distribution on the regression target ensures that at high residuals (between output and ground truth, occurring in the initial learning phase), the gradients are much larger than the standard unimodal Gaussian distribution (explained in detail in Section 3 and Figure 1) leading to faster convergence at the beginning of the learning phase. Second, we also anneal the learning rate over the course of training along with the temperature that avoids gradient explosion towards the end of the learning phase, a problem with the standard heteroscedastic Gaussian-based likelihood distribution with sharp gradients at lower errors. Third, we construct the temperature-dependent likelihood such that the predicted uncertainty is *encouraged* to be calibrated at every step, by being close to the error between the prediction and ground truth.

The standard unimodal distribution faces slow convergence in the beginning and potential gradient explosion towards the end of the learning phase and provides poorly calibrated uncertainty estimates. In contrast, our LIKA method allows faster convergence and offers well-calibrated uncertainty estimates for a broad spectrum of regressions. This also differs from uncertainty regression methods that estimate the full quantile as they are often shown to be effective on low-dimensional regression.

**Contributions.** We introduce a temperature-dependent likelihood annealing scheme for deep regression models with uncertainty estimation that leads to faster model convergence and offers better-calibrated uncertainty (detailed in Section 3.3). We conduct a comprehensive evaluation on various datasets, including chaotic particle trajectory denoising, physical property prediction of molecules using 3D atomistic representation, image super-resolution, and medical image translation using MRI images, presented in Section 4.

## 2 Related Work

Deep neural networks (DNNs) typically estimate inaccurate uncertainty due to their deterministic form that is insufficient for characterizing the accurate confidence (Gal, 2016; Guo et al., 2017). Bayesian inference has been widely studied to effectively estimate uncertainty. Directly performing Bayesian inference on deep nonlinear networks is infeasible due to intractable computations. Hence, approximate inference has been explored by variational inference (Graves, 2011; Blundell et al., 2015; Daxberger et al., 2021; Maddox et al., 2019) or MCMC-based approximation (Welling & Teh, 2011; Chen et al., 2014). However, due to its approximation, the estimated uncertainty may fail to follow the true uncertainty quantification (Lakshminarayanan et al., 2017). Moreover, compared with typical DNNs, approximate Bayesian inference is computationally more expensive and has slower convergence in practice. Non-Bayesian methods have been proposed as an alternative. For instance, (Kendall & Gal, 2017; Lakshminarayanan et al., 2017) modeled two terms, i.e. predictive mean and variance, as an output of DNN to estimate the uncertainty directly from the network's output. Another line of work estimates the uncertainty in the prediction in a non-parametric manner by estimating different quantiles for a given input (Lin et al., 2021; Chen et al., 2021; Zhou et al., 2021; Chung et al., 2021). Moreover, there are also works from conformal predictions that quantify uncertainty by constructing prediction intervals, which are sets of possible outcomes that are believed to contain the true value with a certain probability (Wieslander et al., 2020; Messoudi et al., 2020; Zhang et al., 2021).

In general, there are two broad types of uncertainties in deep learning: (i) Aleatoric and (ii) Epistemic. Aleatoric uncertainty is the uncertainty that arises from the inherent randomness in the data. In contrast, Epistemic uncertainty is the uncertainty that arises due to a lack of knowledge or information about the data. In real-world scenarios with access to large datasets, aleatoric uncertainty is often critical because it is

directly related to the variability in the data, which is essential to modeling real-world scenarios (Monteiro et al., 2020; Mukhoti et al., 2021; Ayhan & Berens, 2018). For example, in medical imaging, different patients may have different degrees of variability in their images due to different factors such as the presence of diseases, body types, or imaging equipment (Wang et al., 2019; Valiuddin et al., 2021; Dohopolski et al., 2020). By modeling aleatoric uncertainty, we can better capture this variability and improve the accuracy of the model. On the other hand, epistemic uncertainty can be reduced by acquiring more data or improving the model architecture (Chen & Techawitthayachinda, 2021; Kendall & Gal, 2017; Swiler et al., 2009). This work focuses on estimating the aleatoric uncertainty in deep regression problems.

Calibrating the inaccurate uncertainty is another way to estimate accurate uncertainty (Guo et al., 2017). In the regression task, calibration was first defined in a quantile manner (Kuleshov et al., 2018). That is, the estimated credible interval with confidence level $\alpha$ (e.g. 95%) is calibrated if $\alpha$% of the ground-truth target is covered in that interval. There are post-processing methods for regression calibration (Kuleshov et al., 2018; Pearce et al., 2018; Tagasovska & Lopez-Paz, 2019). For instance, (Kuleshov et al., 2018) introduced an auxiliary model to adjust the output of the pre-trained model based on Platt-scaling, while others use Gaussian process (Song et al., 2019) or maximum mean discrepancy (Cui et al., 2020). However, an auxiliary model with enough capacity will always be able to recalibrate, even if the predicted uncertainty is completely uncorrelated with the real uncertainty (Laves et al., 2020). Recently, (Levi et al., 2022) extended the definition of calibration where a regressor is well calibrated if the predicted error is equal to the difference between the ground truth and the predicted mean. Using this definition, (Laves et al., 2020) proposed unbiasing the predicted error by optimizing a scaling factor in the post-processing step. However, such methods often add overhead to an already slow model training phase.

## 3 Methodology: Likelihood Annealing

Our framework called Likelihood Annealing (LIKA) belongs to the family of models that are designed to predict a distribution for the outputs (Kendall & Gal, 2017; Laves et al., 2020; Kompa et al., 2021; Upadhyay et al., 2021c;a; 2022) and the model is trained via a loss function derived from maximum likelihood estimation (MLE). We describe the problem formulation and related methods along with their limitations in Section 3.1. We present LIKA that constructs temperature-dependent likelihood to learn faster, better-calibrated regression uncertainty in Section 3.2, and analyze the effects of temperature annealing in Section 3.3.

### 3.1 Background and Motivation

Let $\mathcal{D} = \{(\mathbf{x}_i, \mathbf{y}_i)\}_{i=1}^{i=N}$ be the dataset that comprises of samples from domain $\mathbf{X}$ and $\mathbf{Y}$ (i.e., $\mathbf{x}_i \in \mathbf{X}, \mathbf{y}_i \in \mathbf{Y}, \forall i$), where $\mathbf{X}, \mathbf{Y}$ lies in $\mathbb{R}^m$ and $\mathbb{R}^n$, respectively. The goal of a regression task is to learn a function $\mathbf{\Psi}(\cdot; \theta) : \mathbb{R}^m \to \mathbb{R}^n$ (parameterized by $\theta$) that maps the input $\mathbf{x}$ to the output $\mathbf{y}$. Let $\hat{\mathbf{y}}_i := \mathbf{\Psi}(\mathbf{x}_i; \theta)$ be the estimate for the $\mathbf{y}_i$ and $\epsilon_i := \hat{\mathbf{y}}_i - \mathbf{y}_i$ be the residual between the prediction and the ground-truth. The optimal parameters $(\theta^*)$ are learned by minimizing the error (e.g., $\ell_1$ or $\ell_2$ loss) between the prediction and ground truth using the labeled dataset. The $\ell_1/\ell_2$ loss function to train regression models originate by treating the residuals (i.e., $\epsilon_i$) as following the i.i.d Laplace/Gaussian distribution. However, the i.i.d assumption will not capture the heteroscedasticity, and will allow uncertainty estimation with the limiting assumption of identical, i.e., homoscedastic, uncertainty values.

To estimate the uncertainty, the existing works (Kendall & Gal, 2017) relax the i.i.d assumption and learn to model the heteroscedasticity as well. Such models are learned by maximizing the likelihood. Assuming that residuals follow Gaussian distribution, i.e., $\epsilon_i \sim \mathcal{N}(0, \hat{\sigma}_i)$, the likelihood, $P(\mathcal{D}|\theta)$, is a factored Gaussian distribution, $P(\mathcal{D}|\theta) = \prod_{i=1}^{i=N} \frac{1}{\sqrt{2\pi\hat{\sigma}_i^2}} \exp(-\frac{|\hat{\mathbf{y}}_i - \mathbf{y}_i|^2}{2\hat{\sigma}_i^2})$. the MLE estimates for the parameters are obtained by minimizing the negative-log likelihood,

$$-\log P(\mathcal{D}|\theta) = \sum_{i=1}^{i=N} \frac{\log \hat{\sigma}_i^2}{2} + \frac{|\hat{\mathbf{y}}_i - \mathbf{y}_i|^2}{2\hat{\sigma}_i^2} + Const. \tag{1}$$

The DNN is modified to output both the prediction (i.e., the mean of Gaussian) as well as the uncertainty estimate (i.e., the variance of Gaussian) learned using the above equation, i.e., $\mathbf{\Psi}(\mathbf{x}_i; \theta) = \{\hat{\mathbf{y}}_i, \hat{\sigma}_i\}$. While

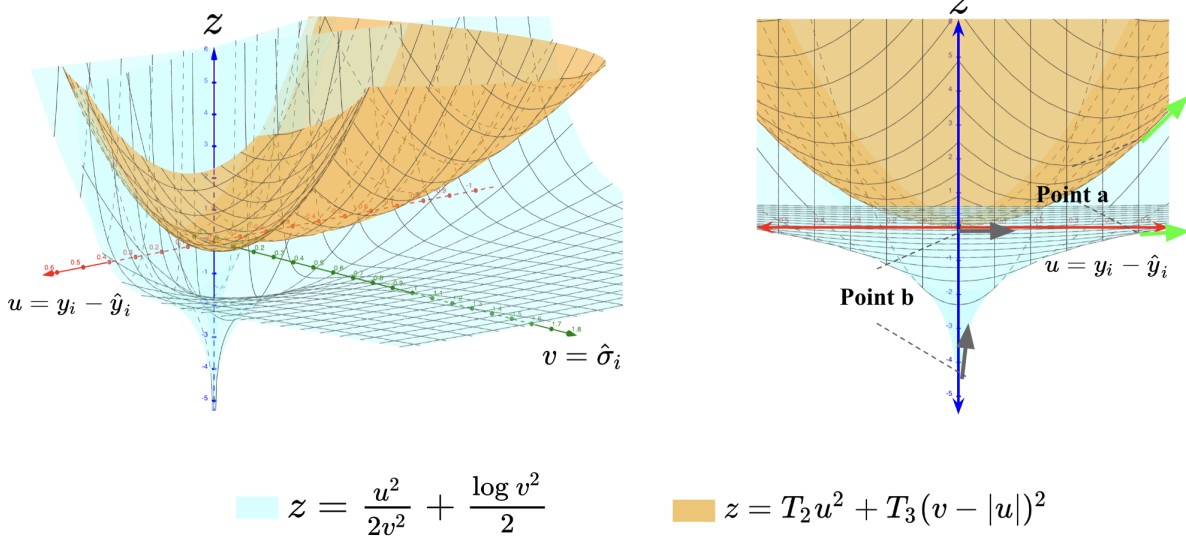

$$z = \frac{u^2}{2v^2} + \frac{\log v^2}{2} \qquad z = T_2 u^2 + T_3(v - |u|)^2$$

Figure 1: (Left) Objective function based on negative log-likelihood of standard heteroscedastic Gaussian distribution (**blue**) and temperature-dependent regularizer (**orange**) from Equation 4 as a function of residual and the estimated standard deviation. (Right) The 2D plot showing surfaces for a fixed predicted variance. The error and predicted variance are high at the beginning of the learning phase. The gradient of the temperature-dependent regularizer is higher (**orange**) than the gradient for the standard objective (**blue**), see **Point a** on both curves. Towards the end of training (with small error, predicted variance, and low temperatures), the objective from Equation 4 is dominated by the negative log-likelihood of standard heteroscedastic Gaussian with non-zero gradients. While gradients from the regularizer are zero, see **Point b**.

this method allows predicting the uncertainty estimates in single forward pass post training, it has several downsides, as discussed in the following. The blue surface in Figure 1-(Left) shows the loss from Equation 1 (which is derived by taking the negative log of Gaussian likelihood). It consists of two variables: the residual $\mathbf{y}_i - \hat{\mathbf{y}}_i$ (denoted by $u$) and the standard deviation $\hat{\sigma}_i$ (denoted by $v$). At the beginning of the training phase, the residual between the prediction and the ground truth is large along with significantly large predicted variance. Still, the corresponding gradient at that point is small (see **Point a** on the blue curve in Figure 1-(Right)), leading to slower convergence towards optima. As the learning progresses, the residual between prediction and ground truth reduces substantially and so does the predicted variance, which leads to very high gradients potentially causing gradient explosion, a phenomenon often observed in practice (see **Point b** on the blue curve in Figure 1-(Left)). Together, this leads to slower model convergence as gradients, in the beginning, are too small. At the same time, the learning rate would also have to be substantially smaller to avoid gradient explosion later. Moreover, the works in (Laves et al., 2020; Levi et al., 2022; Phan et al., 2018) have shown that this method requires an additional post hoc calibration phase to tackle miscalibration.

## 3.2 Constructing Temperature Dependent *Improper* Likelihood

To tackle the slow convergence issue while providing well-calibrated uncertainty estimates, we formulate a temperature-dependent likelihood function that facilitates faster convergence with the help of temperature annealing. Our formulation imposes an explicit condition on the uncertainty estimates, keeping them calibrated throughout the learning phase, leading to calibrated uncertainty estimates without any post-hoc calibration phase. While Equation 1 denotes the negative log-likelihood for the standard Gaussian distribution, We formulate a new improper likelihood distribution on the network output given by,

$$P(\mathcal{D}|\theta) = \prod_{i=1}^{i=N} \frac{e^{\frac{-|\hat{\mathbf{y}}_i - \mathbf{y}_i|^2}{(2\hat{\sigma}_i^2)}}}{\sqrt{2\pi\hat{\sigma}_i^2}} \times e^{-T_2(|\hat{\mathbf{y}}_i - \mathbf{y}_i|^2)} \times e^{-T_3 \left\{ \begin{array}{l} |\hat{\mathbf{y}}_i - (\mathbf{y}_i + \hat{\sigma}_i)|^2, \hat{\mathbf{y}}_i \geq \mathbf{y}_i \\ |\hat{\mathbf{y}}_i - (\mathbf{y}_i - \hat{\sigma}_i)|^2, \hat{\mathbf{y}}_i < \mathbf{y}_i \end{array} \right\}} \tag{2}$$

Where, $T_2, T_3$ are hyper-parameters that we refer to as temperature. We then use the improper maximum likelihood estimator, as also used in (Coretto & Hennig, 2016; 2017; Aghaei et al., 2008) to derive an objective function. We do this by taking the negative log of improper likelihood from Equation 2, leading to the following objective (*omitting the constants for clarity and simplification*):

$$\sum_{i=1}^{i=N} \frac{\log \hat{\sigma}_i^2}{2} + \frac{|\hat{\mathbf{y}}_i - \mathbf{y}_i|^2}{2\hat{\sigma}_i^2} + T_2(|\hat{\mathbf{y}}_i - \mathbf{y}_i|^2) + T_3 \left\{ \begin{array}{l} |\hat{\mathbf{y}}_i - (\mathbf{y}_i + \hat{\sigma}_i)|^2, \hat{\mathbf{y}}_i \geq \mathbf{y}_i \\ |\hat{\mathbf{y}}_i - (\mathbf{y}_i - \hat{\sigma}_i)|^2, \hat{\mathbf{y}}_i < \mathbf{y}_i \end{array} \right\}. \tag{3}$$

We note that the above equation can be re-written as,

$$\sum_{i=1}^{i=N} \frac{\log \hat{\sigma}_i^2}{2} + \frac{|\hat{\mathbf{y}}_i - \mathbf{y}_i|^2}{2\hat{\sigma}_i^2} + T_2(|\hat{\mathbf{y}}_i - \mathbf{y}_i|^2) + T_3(|\hat{\sigma}_i - |\hat{\mathbf{y}}_i - \mathbf{y}_i||^2). \tag{4}$$

Equation 4 has two additional terms (i.e., $T_2(|\hat{\mathbf{y}}_i - \mathbf{y}_i|^2)$ and $T_3(|\hat{\sigma}_i - |\hat{\mathbf{y}}_i - \mathbf{y}_i||^2)$) compared to Equation 1. To understand the effects of our proposed temperature-dependent improper likelihood, we first, look at the newly introduced temperature-dependent regularizers, represented by $\mathcal{L}_{\text{reg}}$ given by,

$$\mathcal{L}_{\text{reg}} = T_2(|\hat{\mathbf{y}} - \mathbf{y}|^2) + T_3(|\hat{\sigma} - |\hat{\mathbf{y}} - \mathbf{y}||^2). \tag{5}$$

Figure 1-(Left) shows the surface corresponding to $\mathcal{L}_{\text{reg}}$ in **orange** for substantially large temperature values. We notice that at the beginning of the training phase, with temperature hyper-parameters set to high values, Equation 4 is dominated by $\mathcal{L}_{\text{reg}}$. As shown in Figure 1-(Right), the corresponding gradient at the beginning of the training (dominated by $\mathcal{L}_{\text{reg}}$) is much higher (see **Point a** on the orange curve). This encourages faster convergence at the beginning of the training phase, unlike the Equation 1.

To further understand the effects of the newly introduced temperature-dependent regularizers, we look at the conceptual schematic, shown in Figure 2, that illustrates the soft constraint imposed by the regularizers, represented by $\mathcal{L}_{\text{reg}}$. As discussed above, we propose to start with high values for the $T_2$ and $T_3$ hyper-parameters and gradually decrease them during the course of training. We observe that at high temperatures (i.e., at the beginning of the training phase), the objective function from Equation 4 is dominated by the last two terms that are controlled by $T_2$ and $T_3$. We show these two terms (i.e., $\mathcal{L}_{\text{reg}}$) in Figure 2 as a function of $\hat{\mathbf{y}}$ for a fixed ground truth $\mathbf{y}$ and a fixed $\hat{\sigma}$, and note that minimizing $\mathcal{L}_{\text{reg}}$ encourages the prediction $\hat{\mathbf{y}}$ to be close to the ground truth $\mathbf{y}$, while also ensuring that the discrepancy between the prediction and ground truth $|\hat{\mathbf{y}} - \mathbf{y}|^2$ is close to the predicted variance $\hat{\sigma}^2$, encouraging calibration of the predicted variance without the need of post-hoc techniques (orange **bold** curve in Figure 2).

Moreover, as the training progresses and the temperature decreases, $\hat{\mathbf{y}}$ comes closer to $\mathbf{y}$ and the predicted variance $\hat{\sigma}$ also decreases, we notice that this leads to the local optimums com-

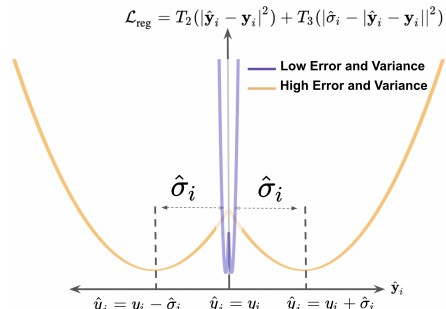

Figure 2: Schematic of the temperature-dependent regularizer characterized by $\{\mathbf{y}, \hat{\mathbf{y}}, \hat{\sigma}\}$. This enforces the prediction to be close to ground truth and the uncertainty estimate to be close to the error, i.e., calibrated (shown in **orange**). When the predicted variance is small, all the optimums come close to each other (shown in **blue**).

ing closer, and eventually collapsing at $\hat{\mathbf{y}} = \mathbf{y}$ in the limit (blue **bold** curve in Figure 2). Throughout the early phase of training (with high temperature), the regularizer encourages the prediction $\hat{\mathbf{y}}$ to be close to ground truth $\mathbf{y}$ and the predicted variance $\hat{\sigma}^2$ to be close to error $|\hat{\mathbf{y}} - \mathbf{y}|^2$. This way, the regularizer imposes a soft constraint for calibration of the predicted uncertainty estimate throughout the training.

### 3.3 Effects of Temperature Annealing

The temperature-dependent improper likelihood in Equation 2 leads to objective in Equation 4 that allows us to control the contribution of individual terms by changing the temperature hyper-parameters $T_2, T_3$.

As described in Section 3.2, annealing the temperature hyperparameters allow faster convergence of the uncertainty-aware regression with better-calibrated uncertainty methods. We start by initializing $T_2, T_3$ with a high value of 100 and progressively reduce them according to the training epochs using exponential annealing – referred to as *temperature annealing*. At higher temperatures, the overall objective is dominated by the

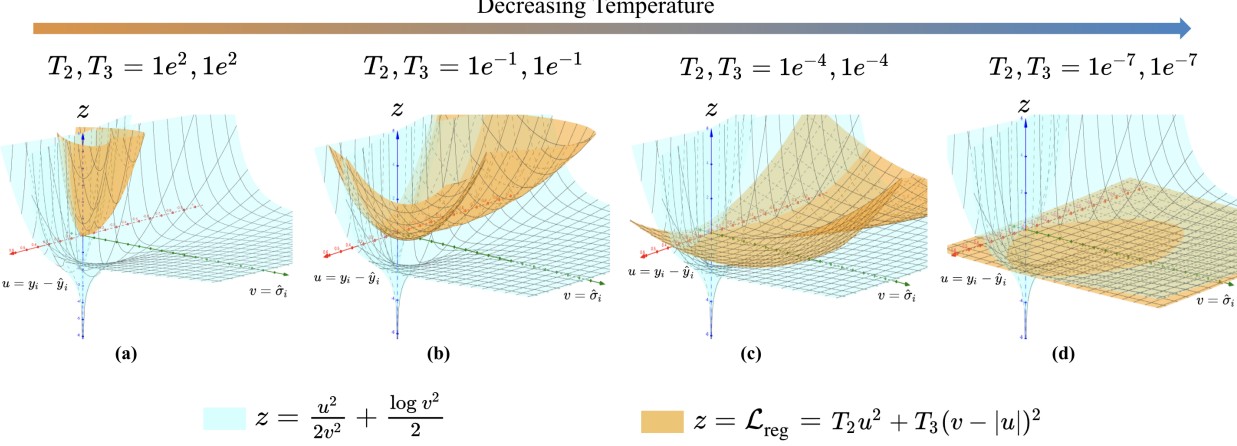

Figure 3: Effects of temperature annealing. As we anneal the temperature in Equation 4, the proposed temperature-dependent regularizer $\mathcal{L}_{\text{reg}}$ from Equation 5 (shown in **orange**) gradually changes from (a), (b), (c) to (d), which provides faster convergence at the beginning of training while ensuring convergence to the same optima as the standard objective function as described in Equation 1 (shown in **blue**).

temperature-dependent terms ($\mathcal{L}_{\text{reg}}$). Figure 3-(a)) shows the loss surface for the negative log-likelihood derived from standard Gaussian (i.e., Equation 1) and the newly introduced temperature-based regularize $\mathcal{L}_{\text{reg}}$. As the temperatures decrease, the overall loss is close to the standard loss function. This can also be seen from Figure 3-(b,c), where the surface corresponding to $\mathcal{L}_{\text{reg}}$ flattens out at lower temperature, eventually coming close to plane surface as temperatures approach 0 as shown in Figure 3-(d)). Note that, when temperatures are zero $\mathcal{L}_{\text{reg}} = 0$ and Equation 4 reduces to Equation 1. This dynamic contribution from different terms allows the network to converge faster in the beginning (as gradients from the temperature-dependent loss terms are higher than the standard loss term), and ensures stable convergence to the same optima as the standard loss, thus leading to faster, better-calibrated uncertainty.

## 3.4 Normalizing the improper Likelihood

We further study our proposed improper likelihood (presented at Equation 2) to convert it into proper likelihood. This is achieved by normalizing Equation 2. Let the normalizing constant be $Z_i$. Then the proper likelihood is given by,

$$P(\mathcal{D}|\theta) = \prod_{i=1}^{i=N} Z_i e^{\frac{-|\hat{\mathbf{y}}_i - \mathbf{y}_i|^2}{(2\hat{\sigma}_i^2)}} \times e^{-T_2(|\hat{\mathbf{y}}_i - \mathbf{y}_i|^2)} \times e^{-T_3 \left\{ \begin{array}{l} |\hat{\mathbf{y}}_i - (\mathbf{y}_i + \hat{\sigma}_i)|^2, \hat{\mathbf{y}}_i \geq \mathbf{y}_i \\ |\hat{\mathbf{y}}_i - (\mathbf{y}_i - \hat{\sigma}_i)|^2, \hat{\mathbf{y}}_i < \mathbf{y}_i \end{array} \right\}} \tag{6}$$

In the above, $Z_i = \dfrac{2\sqrt{\pi}\hat{\sigma} \exp\left(-\frac{\hat{\sigma}^2 T_3\left(2\hat{\sigma}^2 T_2 + 1\right)}{2\hat{\sigma}^2(T_2 + T_3) + 1}\right)\left(\text{erf}\left(\frac{2\hat{\sigma}^2 T_3}{\sqrt{4\hat{\sigma}^2(T_2+T_3)+2}}\right)+1\right)}{\sqrt{4\hat{\sigma}^2(T_2+T_3)+2}}$. The negative log-likelihood of Equation 6 leads to following objective,

$$\mathcal{L}_{\text{norm}} = \sum_{i=1}^{i=N} -\left(\frac{\hat{\sigma}_i^2 T_3\left(2\hat{\sigma}_i^2 T_2 + 1\right)}{2\hat{\sigma}_i^2\left(T_2 + T_3\right) + 1}\right) + \log\left(\text{erf}\left(\frac{2\hat{\sigma}_i^2 T_3}{\sqrt{4\hat{\sigma}_i^2\left(T_2 + T_3\right) + 2}}\right) + 1\right) - 0.5\log\hat{\sigma}_i^2 + \frac{|\hat{\mathbf{y}}_i - \mathbf{y}_i|^2}{2\hat{\sigma}_i^2}$$
$$+ T_2(|\hat{\mathbf{y}}_i - \mathbf{y}_i|^2) + T_3(|\hat{\sigma}_i - |\hat{\mathbf{y}}_i - \mathbf{y}_i||^2) \tag{7}$$

## 4 Experiments

We first provide a detailed description of our experimental setup, including the datasets used for training and evaluation, the evaluation metrics employed to assess the performance of our model in Section 4.1. We compare our model to a wide variety of state-of-the-art methods quantitatively and qualitatively in Section 4.2. Finally, we also provide an ablation analysis in Section 4.2 to study the rationale of our model formulation.

### 4.1 Experimental Setup

**Datasets and Tasks.** We conduct experiments on five datasets (three small scale problems, two large scale problems) to solve the regression task and provide uncertainty estimation.

We choose the following three low-dimensional regression problems. They highlight the different complexities and network architectures that are required to solve them. In *Chaotic System using Lorenz Attractor* (referred to as *Lorenz Attractor*), the Lorenz equations describe non-linear chaotic systems given by, $\frac{\partial z_1}{\partial t} = 10(z_2 - z_1)$, $\frac{\partial z_2}{\partial t} = z_1(28 - z_3) - z_2$, $\frac{\partial z_3}{\partial t} = z_1 z_2 - 8z_3/3$. Similar to (Garcia Satorras et al., 2019), to generate a trajectory we run the Lorenz equations with a $\partial t = 10^{-5}$ from which we sample with a time step of $t = 0.05$. Each point is then perturbed with Gaussian noise of standard deviation 0.5 to produce pairs of noisy and clean trajectories split into non-overlapping train/validation/test sets. We use a 1D CNN to map the noisy input to clean output. The *Physical Properties of Molecules (Atom3D)* (Townshend et al.) is a 3D molecular structure dataset aiming to predict the physical property such as the dipole moment given the 3D atomistic representation. We use the standard Graph Neural Network (GNN) for this task. The *House Price Prediction (Boston-housing)* (Harrison Jr & Rubinfeld, 1978; Belsley et al., 2005) dataset is used to predict the house prices using various attributes using Multi Layer Perceptrons (MLPs).

To show the generalization of our method to high-dimensional regression problems, we use the following two datasets. In *Super-resolution of Natural Images (Super-resolution)*, we learn mapping from low-resolution to high-resolution images using CNNs, using DIV2K dataset (Timofte et al., 2018; Ignatov et al., 2019). We do 4x downsampling to create the corresponding low-resolution images. The dataset is split into 800/100/100 images for training/val/test sets. In *Medical Image Translation (MRI Translation)*, We translate one imaging modality to another, i.e., T1 MRI to T2 MRI images. As T1 and T2 MRI from the same patient in the same orientation are often not available and T2 takes longer to acquire, learning a mapping from T1 to T2 is desirable. As in (Upadhyay et al., 2021a), we use T1 and T2 MRI of 500 patients from IXI dataset (Robinson et al., 2010) (200/100/200 for training/val/test) in a 2D CNN based on U-Net (Ronneberger et al., 2015).

**Evaluation Metrics.** To measure the quality of regression output, we adopt the standard metrics: mean absolute error (`MAE`) and mean square error (`MSE`). In addition, for the super-resolution and medical image translation tasks, we use `PSNR` and `SSIM` to measure the structural similarity between two images (Wang et al., 2004). To measure the quality of uncertainty estimates ($\hat{\sigma}^2$), we compute (i) the correlation coefficient (`Corr. Coeff.`) between uncertainty estimates ($\hat{\sigma}^2$) and the error ($|\hat{\mathbf{y}} - \mathbf{y}|^2$). (ii) *Uncertainty calibration error* (`UCE`) for regression tasks (Laves et al., 2020; Levi et al., 2022). Following (Guo et al., 2017), the uncertainty output $\hat{\sigma}^2$ of a deep model is partitioned into $M$ bins with equal width (each represented by $B_m$ for $\forall m \in \{1, 2...M\}$). A weighted average of the difference between the predictive error and uncertainty is used, $\text{UCE} = \sum_{m=1}^{M} \frac{|B_m|}{N} |\text{err}(B_m) - \text{uncer}(B_m)|$. Where, $\text{err}(B_m) := \frac{1}{|B_m|} \sum_{i \in B_m} ||\hat{\mathbf{y}}_i - \mathbf{y}_i||^2$ and $\text{uncer}(B_m) := \frac{1}{|B_m|} \sum_{i \in B_m} \hat{\sigma}_i^2$. (iii) UCE for the re-calibrated uncertainty estimates (`R.UCE`). We use post-hoc calibration technique introduced in (Laves et al., 2020), called $\sigma$-scaling, that optimizes for the scaling factor ($s$), post training to produce uncertainty estimates ($\hat{\sigma}^2$) and predictions ($\hat{\mathbf{y}}$) using, $s^* = \underset{s}{\text{argmin}} \left[ N \log(s) + \frac{1}{2s^2} \sum_{i=1}^{N} \frac{|\hat{\mathbf{y}}_i - \mathbf{y}_i|^2}{\hat{\sigma}_i^2} \right]$. In addition, we present the (iv) *expected calibration error* (`ECE`) and (v) *sharpness* (`Sharpness`). While ECE is another metric to quantify the calibration of the uncertainty estimates, one must note that it may be possible to have an uninformative, yet average calibrated model (Chung et al., 2021; Zhou et al., 2021). Therefore it is necessary to also present the Sharpness metric that encourages more-concentrated distributions. Finally, we present the (vi) predictive log-likelihood that assesses how well the predicted conditional distribution fits the data.

**Implementation Details.** Our LIKA method is generalizable across different types of architectures. Here we perform experiments with MLPs, 1D/2D CNNs, and GNNs. We take the well-established networks for

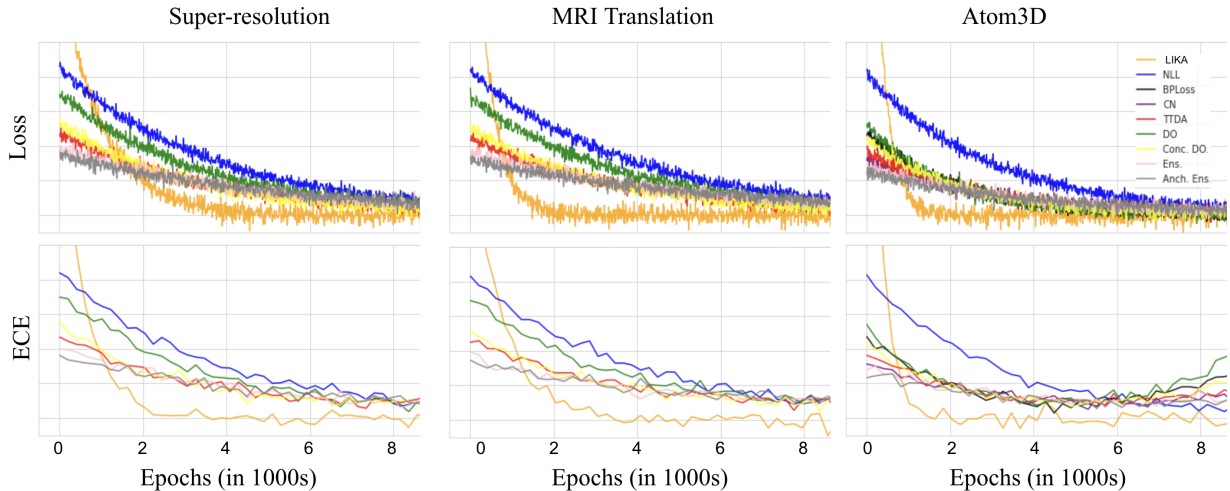

Figure 4: Plots comparing the required convergence time (number of epochs to converge) for different methods and corresponding ECE during the training on (i) Super-resolution, (ii) MRI translation, (iii) Atom3D.

the respective problems and modify them to produce the uncertainty estimates as described in (Kendall & Gal, 2017; Sudarshan et al., 2021). All the networks were trained using Adam optimizer (Kingma & Ba, 2014). The initial learning rate was set to $2e^{-4}$ and cosine annealing was used to decay the learning rate over the course of the learning phase. The hyper-parameters, $(T_2, T_3)$ (Equation 4) were set to $(100, 100)$ and scheduled to exponentially decay over the course of the training. We provide the code in the supplementary.

## 4.2 Comparing to Uncertainty Estimation Methods

**Compared methods.** For each of the regression tasks, we compare our model (LIKA) to eight representative state-of-the-art methods for uncertainty estimation using DNNs for regression tasks, belonging to a diverse class of methods, i.e. Bayesian ensemble, test-time data augmentation, maximum likelihood and variants of the same, and finally quantile regression methods. In addition, we evaluate LIKA-Norm for some of our experiments. This method uses proper likelihood-based objective to train the network given by Equation 6.

*Bayesian methods:* In (DO) (Gal & Ghahramani, 2016) the weights of the neural network are randomly dropped at training and inference time. Multiple forward passes for the same input at inference time allow us to estimate the uncertainty. In Concrete Dropout (Conc. DO.) (Gal et al., 2017) the optimal dropout probability for the weights of the neural network is learned at training. While the above methods only consider the epistemic uncertainty, we also evaluate DO-NLL, which is similar to DO, except the head is split into two to predict both the mean and variances using the Gaussian-NLL loss function, along with dropouts during training and evaluation for *Boston Housing* and *Atoms3D* dataset. For DO-NLL, we consider the aleatoric uncertainty for evaluation obtained as the mean of variance head outputs at evaluation for a single sample with 100 forward passes and dropouts activated.

*Ensemble Methods:* In Deep Ensemble (Ens) (Lakshminarayanan et al., 2017) multiple deterministic networks are trained to make the final prediction with uncertainty estimates. In Anchored Ensemble (Anch. Ens.) (Pearce et al., 2020) the weights of the neural networks in the ensemble are regularized about values drawn from a prior distribution, allowing approximate Bayesian inference. While the above estimates epistemic uncertainty, to capture the aleatoric uncertainty, We also evaluate Ens-NLL for *Boston Housing* and *Atoms3D* dataset, which is an ensemble of 5 similar models except for the head of each model in the ensemble is split into two to predict both the mean and variances using the Gaussian-NLL loss. Each ensemble model is trained independently, with different weight parameter initializations. The aleatoric uncertainty considered in evaluation of Ens-NLL is the mean of variance head for all the models in the ensemble.

*Test-time Data Augmentation Methods:* In Test Time Data Augmentation (TTDA) (Wang et al., 2019; Ayhan & Berens, 2018; Gawlikowski et al., 2021) multiple perturbed copies of the input are passed through a deterministic network to estimate the predictive uncertainty at the inference stage.

*Maximum likelihood methods:* In this method (NLL) (Kendall & Gal, 2017; Sudarshan et al., 2021) the network is modified to predict the mean and variance and then trained by optimizing negative log-likelihood. The variance head then provides uncertainty estimates for the prediction at the inference time. For *Boston Housing* and *Atoms3D* data, we also evaluate (Stirn et al., 2022) (called NLL-FH) that uses a modified objective instead of the NLL of heteroscedastic Gaussian, using a backbone architecture similar to NLL (and other methods in this work), with the head split to predict both mean and variance as (Kendall & Gal, 2017).

*Quantile Regression Methods:* In Calibrated Quantile Regression Method (BPLoss) (Chung et al., 2021) proposes a model that specifies the full quantile function for the predictions and achieves a balance between calibration and sharpness. In Collaborating Networks for estimating uncertainty intervals (CN) (Zhou et al., 2021) two networks are trained simultaneously, one to estimate the cumulative distribution function, and the other approximates its inverse. We note that some baseline methods (i.e., BPLoss and CN) have only been proposed for low-dimensional regression settings (where the output of a model is single scalar) and it is non-trivial and inefficient to scale it to high-dimensional regression settings (e.g., image translation, where the output for an input is a high-dimensional matrix/tensor). Therefore such models are compared only on low-dimensional regression tasks where they are applicable.

**Quantitative results on convergence.** In this experiment, we train different models to perform the different kinds of regression task and keep track of the training and validation loss to identify if the model has converged. For all the models we used the same optimizer (i.e., Adam (Kingma & Ba, 2014)) with the same initial learning rate (i.e., $\mathtt{lr} = 2e^{-4}$) and identical decaying schedule (i.e., cosine annealing for $\mathtt{lr}$).

We observe in Figure 4 that the baseline methods consistently take longer time to converge while our proposed method (LIKA) consistently has faster convergence. For instance, on the super-resolution task, our method takes about 4,000 epochs to converge while the other baseline methods consistently take longer than 8000 epochs to converge. In particular, the NLL baseline takes the longest to converge. We also note that in the early phase of training, our LIKA has much higher loss, this is due to the additional temperature dependent loss terms (in Equation 4) that contribute to the overall loss. However, the higher values of the temperature $T_2$ and $T_3$ in the beginning of the training phase also allow faster convergence, as explained in Section 3. Moreover, towards the end of the training phase, the temperature parameters are annealed to a low value (close to zero) and the over all loss function reduces to a low value.

Figure 4 (second row) shows the evolution of ECE for the derived uncertainty using various methods during the training. Again we see that our LIKA achieves the lowest ECE much faster than the other methods. A similar trend is observed for the other datasets. For example, on Atom3D dataset, the proposed method converges at about 2000 epochs, much faster than other baselines, similarly, it achieves the lowest ECE much faster than other methods. These results show that our method converges much faster than the other methods, which is in line with our motivation to ensure a faster convergence for the regression uncertainty model along with better-calibrated uncertainty as described in Section 3.3.

**Quantitative results on regression and uncertainty.** Uncertainty-aware regression models must be evaluated on two fronts which are (i) the regression performance, i.e., the quality of the target predictions and (ii) the quality of estimated uncertainty (the uncertainty should be sharp and well calibrated). We evaluate the model performance based on two set of metrics: (1) task-specific metrics that evaluate the regression results using `MAE`, `MSE`, `PSNR`, `SSIM`, and (2) calibration-specific metrics that evaluate the quality of the uncertainty estimates using `C.Coeff.`, `UCE`, `R.UCE`, `ECE`, `Sharp.`, and `Log-likeli.` Table 1 shows the quantitative results that evaluate regression and the quality of uncertainty estimates for different methods on multiple regression tasks. Our LIKA method also obtains high quality regression outputs. In two tasks (including super-resolution, and MRI translation), our LIKA achieves the best or competitive performance compared to the other methods. We note that while no single metric can indicate the "goodness" of uncertainty estimates (as there is no groundtruth for uncertainty values), the collective set of metrics such as `C.Coeff.`, `UCE`, `R.UCE`, `ECE`, `Log-likeli`, `Sharp.` provide a holisitic indication of "goodness" of uncertainty metric. The proposed method, LIKA, consistently performs well in terms of the above metrics. Overall, these quantitative results show that our method performs well in providing both satisfactory regression and uncertainty estimates.

**Qualitative results on regression and uncertainty.** Figure 5 shows the regression output on different datasets. Figure 5-(i) & (ii) visualizes the generated images for image super-resolution and MRI translation

| T | Methods | Metrics | | | | | | | | | |
|---|---|---|---|---|---|---|---|---|---|---|---|
| | | MAE ↓ | MSE ↓ | SSIM ↑ | PSNR ↑ | C.Coeff. ↑ | UCE↓ | R.UCE↓ | Log-likeli. ↑ | ECE ↓ | Sharp. ↓ |
| Boston-housing | DO (Gal & Ghahramani, 2016) | 2.851 | 13.26 | - | - | 0.014 | 10.76 | 10.18 | -2.46 | 10.2 | 8.66 |
| | DO-NLL (Gal & Ghahramani, 2016) | 2.661 | 12.83 | - | - | 0.116 | 8.783 | 8.226 | -2.21 | 9.32 | 8.48 |
| | Conc. DO (Gal et al., 2017) | **2.413** | 10.18 | - | - | 0.135 | 9.882 | 9.126 | -2.15 | 9.18 | 9.16 |
| | Ens (Lakshminarayanan et al., 2017) | 2.971 | 13.76 | - | - | 0.011 | 11.26 | 10.78 | -2.41 | 10.3 | 8.87 |
| | Ens-NLL (Lakshminarayanan et al., 2017) | 2.913 | 12.82 | - | - | 0.114 | 10.38 | 9.98 | -2.33 | 10.1 | 8.27 |
| | Anch. Ens. (Pearce et al., 2020) | 2.553 | **10.11** | - | - | 0.154 | 9.547 | 9.135 | -2.32 | 9.92 | 9.82 |
| | TTDA (Wang et al., 2019; Ayhan & Berens, 2018; Gawlikowski et al., 2021) | 2.584 | 10.30 | - | - | 0.007 | 14.32 | 13.85 | -2.24 | 11.8 | 9.28 |
| | NLL (Kendall & Gal, 2017) | 2.663 | 10.75 | - | - | 0.107 | 12.67 | 12.23 | -2.42 | 11.5 | 8.21 |
| | NLL-FH (Stirn et al., 2022) | 2.551 | 10.14 | - | - | 0.103 | 13.63 | 13.11 | -2.62 | 11.7 | 8.33 |
| | BPLoss (Chung et al., 2021) | 2.684 | 11.49 | - | - | 0.237 | 9.216 | 8.837 | -2.11 | 9.72 | 9.01 |
| | CN (Zhou et al., 2021) | 2.594 | 11.13 | - | - | 0.213 | 10.84 | 9.722 | -2.23 | 9.65 | 9.67 |
| | LIKA (ours) | 2.593 | 10.51 | - | - | **0.348** | **0.756** | **0.637** | **-2.06** | **6.37** | **8.22** |
| | LIKA-Norm (ours) | 2.633 | 10.94 | - | - | 0.311 | 0.818 | 0.682 | -2.08 | 6.87 | 9.14 |
| Atom3D | DO (Gal & Ghahramani, 2016) | 1.950 | 5.828 | - | - | 0.085 | 5.380 | 5.054 | -0.24 | 2.12 | 4.32 |
| | DO-NLL (Gal & Ghahramani, 2016) | 0.950 | 1.224 | - | - | 0.135 | 4.177 | 3.956 | -0.22 | 1.92 | 4.11 |
| | Conc. DO (Gal et al., 2017) | 1.834 | 5.212 | - | - | 0.136 | 4.879 | 4.122 | -0.21 | 1.81 | 4.18 |
| | Ens (Lakshminarayanan et al., 2017) | 1.215 | 2.388 | - | - | 0.138 | 4.623 | 4.376 | -0.23 | 1.69 | 4.17 |
| | Ens-NLL (Lakshminarayanan et al., 2017) | 0.922 | 0.983 | - | - | 0.166 | 4.115 | 3.977 | -0.22 | 1.66 | 4.02 |
| | Anch. Ens. (Pearce et al., 2020) | 1.087 | 1.743 | - | - | 0.182 | 4.124 | 3.763 | -0.26 | 1.42 | 3.95 |
| | TTDA (Wang et al., 2019; Ayhan & Berens, 2018; Gawlikowski et al., 2021) | 0.903 | 1.301 | - | - | 0.157 | 4.167 | 3.988 | -0.38 | 1.94 | 4.78 |
| | NLL (Kendall & Gal, 2017) | **0.498** | **0.463** | - | - | 0.164 | 3.358 | 3.335 | -0.22 | 1.38 | 3.32 |
| | NLL-FH (Stirn et al., 2022) | 0.507 | 0.582 | - | - | 0.112 | 4.468 | 4.112 | -0.32 | 2.24 | 3.82 |
| | BPLoss (Chung et al., 2021) | 0.527 | 0.873 | - | - | 0.189 | 3.527 | 3.166 | -0.21 | 1.55 | **3.12** |
| | CN (Zhou et al., 2021) | 0.521 | 0.845 | - | - | 0.087 | 4.311 | 2.971 | **-0.16** | 1.77 | 3.18 |
| | LIKA (ours) | 0.513 | 0.495 | - | - | **0.567** | **0.296** | **0.277** | -0.18 | **1.37** | 3.17 |
| | LIKA-Norm (ours) | 0.554 | 0.585 | - | - | 0.511 | 0.377 | 0.315 | -0.20 | 1.68 | 3.92 |
| Lorenz Attractor | DO (Gal & Ghahramani, 2016) | 1.373 | 3.463 | - | 29.85 | 0.281 | 2.864 | 2.134 | -0.16 | 4.34 | 5.67 |
| | Conc. DO (Gal et al., 2017) | 1.247 | 3.198 | - | 30.34 | 0.311 | 2.379 | 2.136 | -0.14 | 4.13 | **5.22** |
| | Ens. (Lakshminarayanan et al., 2017) | 2.544 | 11.65 | - | 24.32 | 0.778 | 6.726 | 6.294 | -0.22 | 10.4 | 8.43 |
| | Anch. Ens. (Pearce et al., 2020) | 2.122 | 10.12 | - | 25.64 | 0.432 | 8.756 | 8.154 | -0.29 | 10.7 | 9.43 |
| | TTDA (Wang et al., 2019; Ayhan & Berens, 2018; Gawlikowski et al., 2021) | 1.391 | 3.764 | - | 29.16 | 0.438 | 3.325 | 3.077 | -0.17 | 5.96 | 8.91 |
| | NLL (Kendall & Gal, 2017) | 0.172 | 0.048 | - | 31.28 | 0.588 | 2.368 | 1.933 | -0.13 | **4.33** | 7.87 |
| | LIKA (ours) | **0.153** | **0.029** | - | **32.33** | **0.821** | **0.779** | **0.356** | **-0.11** | 4.36 | 9.12 |
| Super-resolution | DO (Gal & Ghahramani, 2016) | 0.832 | 0.548 | 0.947 | 35.64 | 0.033 | 0.748 | 0.519 | -0.38 | 4.67 | 6.32 |
| | Conc. DO (Gal et al., 2017) | 0.801 | 0.423 | 0.951 | 35.71 | 0.134 | 0.711 | 0.494 | -0.36 | 4.43 | 8.82 |
| | Ens. (Lakshminarayanan et al., 2017) | 0.793 | 0.462 | 0.953 | 36.61 | 0.029 | 0.941 | 0.733 | -0.36 | 8.76 | 10.2 |
| | Anch. Ens. (Pearce et al., 2020) | 0.755 | 0.441 | 0.957 | 36.63 | 0.178 | 0.883 | 0.713 | -0.41 | 8.11 | 9.21 |
| | TTDA (Wang et al., 2019; Ayhan & Berens, 2018; Gawlikowski et al., 2021) | 0.883 | 0.691 | 0.939 | 34.94 | 0.047 | 1.175 | 0.994 | -0.39 | 11.3 | 10.3 |
| | NLL (Kendall & Gal, 2017) | 0.693 | 0.414 | 0.955 | 37.15 | 0.189 | 0.581 | 0.512 | -0.36 | 1.45 | 2.73 |
| | LIKA (ours) | **0.618** | **0.351** | **0.962** | **37.87** | **0.518** | **0.104** | **0.053** | **-0.16** | **0.74** | **0.83** |
| MRI Translation | DO (Gal & Ghahramani, 2016) | 0.732 | 0.683 | 0.912 | 32.45 | 0.159 | 0.864 | 0.771 | -0.33 | 4.48 | 6.23 |
| | Conc. DO (Gal et al., 2017) | 0.715 | 0.612 | 0.917 | 32.97 | 0.189 | 1.125 | 0.932 | -0.31 | 4.12 | 7.89 |
| | Ens. (Lakshminarayanan et al., 2017) | 0.681 | 0.611 | 0.927 | 33.76 | 0.110 | 1.143 | 0.974 | -0.36 | 4.86 | 7.21 |
| | Anch. Ens. (Pearce et al., 2020) | 0.655 | **0.532** | 0.933 | 33.84 | 0.166 | 1.122 | 0.913 | -0.34 | 5.88 | 7.32 |
| | TTDA (Wang et al., 2019; Ayhan & Berens, 2018; Gawlikowski et al., 2021) | 0.755 | 0.729 | 0.904 | 32.18 | 0.128 | 1.483 | 1.153 | -0.37 | 7.21 | 9.74 |
| | NLL (Kendall & Gal, 2017) | 0.632 | 0.582 | 0.938 | 34.34 | 0.134 | 1.673 | 1.448 | **-0.28** | 4.03 | **5.12** |
| | LIKA (ours) | **0.615** | 0.537 | **0.946** | **35.27** | **0.432** | **0.098** | **0.062** | -0.30 | **3.26** | 5.78 |

Table 1: Evaluating different methods on five datasets using `MAE`, `MSE`, `PSNR`, `SSIM` (where applicable, to evaluate regression) and `C.Coeff.`, `UCE`, `R.UCE`, `Log-Likeli.`, `ECE`, `Sharp.` (to measure quality of uncertainty estimates). ↑/↓ indicates higher/lower is better. "T": tasks. Best results are in **bold**.

tasks. While the other methods often generate relatively blurry images with artifacts in colours, our model produces better output visually more similar to the ground-truth. Moreover, Figure 5-(i) & (ii) also shows the uncertainty maps, along with the prediction and error for super-resolution and MRI translation. We observe that for compared methods, uncertainty maps do not always agree with error maps at pixel level (i.e., higher/lower uncertainty than the corresponding error), whereas our uncertainty maps are in agreement with the errors. This suggests that our model provides better-calibrated uncertainty. Figure 5-(iii) shows the plots for predictions vs ground-truth on the Atom3D dataset. We can see that compared to other methods, our method yields predictions much closer to the ground-truth e.g., on the Atom3D dataset, our method produces regression output more highly correlated with the ground-truth. Figure 5-(iv) shows the input noisy trajectory, denoised output and the corresponding ground-truth for the Lorentz attractor dataset. We can see that compared to other methods, our method yields smoother trajectories.

## 4.3 Ablation Analysis of Annealing

Table 2 shows the ablation study of two temperature hyperparameters in our formulated temperature-dependent likelihood (Equation 2) along with different choices of priors for the super-resolution task.

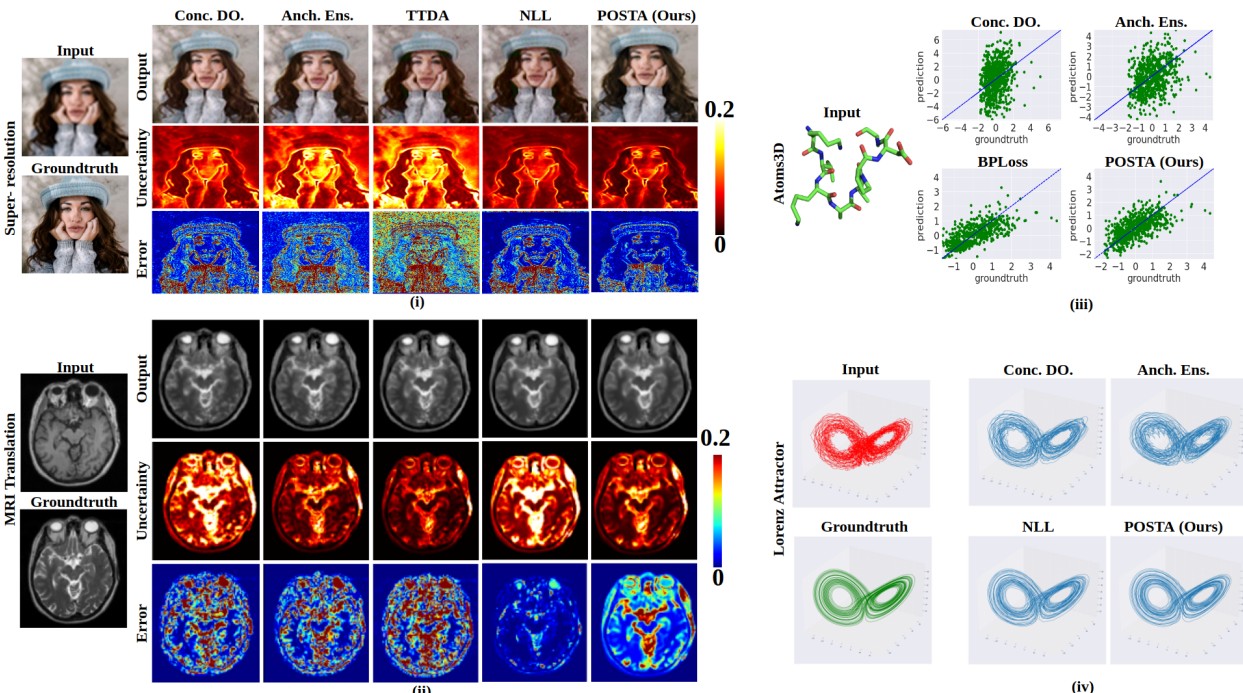

Figure 5: Qualitative results: input, predictions, groundtruth, and the error.

We test the baseline that removes both temperature-dependent terms (i.e. $T_2 = T_3 = 0$) with a uniform prior, this is equivalent to the NLL method and is shown in the first row (`MAE` of 0.693). We then study the effect of fixing one of the temperatures at a non-zero value while setting the other temperature to 0. With $T_2 = 100, T_3 = 0$, we see a slight improvement in regression performance (`MAE` of 0.614 vs. 0.693) and much poorer performance with respect to uncertainty calibration (`UCE` of 1.169 vs. 0.581), this is due to more weighting of fidelity term between the prediction and the ground-truth along with suppression of the default calibration effect of NLL. On the other hand, $T_2 = 0, T_3 = 100$ suppresses the default fidelity term for NLL, therefore the output is of significantly worse quality (poor regression scores, `MAE` of 1.395 vs 0.693) this further degrades the quality of the uncertainty estimates (`UCE` 3.733 vs 0.581). We notice that if the model does not perform good regression, the quality of uncertainty estimate is also adversely effected.

We then study the effects of decaying one of the temperatures while setting other to 0. With $T_2$ decaying (i.e., $T_2 = \downarrow, T_3 = 0$) we see slightly better performance than $T_2 = 100, T_3 = 0$ (`MAE` of 0.612 vs. 0.614 and `UCE` of 0.983 vs. 1.169), whereas with $T_2$ decaying (i.e., $T_2 = 0, T_3 = \downarrow$) we see good regression performance but also an improved calibration performance (`UCE` of 0.152 vs. 0.581). With both the parameters decaying (i.e., $T_2 = \downarrow, T_3 = \downarrow$) we achieve improved regression and calibration results concluding that annealing works the best. In addition to uniform prior setup (i.e., $P(\theta) = \mathcal{U}(\theta)$), we evaluate two other priors (i) Gaussian prior on the parameters of the network, i.e., $P(\theta) = \mathcal{N}(\theta)$ that is equivalent to $\ell_2$ regularization of weights and (ii) Laplace prior, i.e., $P(\theta) = \mathcal{E}(\theta)$ that is equivalent to $\ell_1$ regularization of weights. With Gaussian/Laplace prior we achieve `MAE` of 0.625/0.612 showing that carefully crafted priors may further boost the performance, designing such priors will be explored in future works.

## 4.4 Evaluation on Out-of-Distribution Data

Previous works have studied the performance of various uncertainty-aware methods in the presence of out-of-distribution (OOD) samples at the inference time (Ovadia et al., 2019; Hendrycks et al., 2019; Nandy et al., 2020; Mundt et al., 2019). To evaluate if better quality of uncertainty estimates lead to better OOD performance, we evaluate all the uncertainty trained model for MRI Translation on OOD samples. MRI image acquisition is a noisy process that leads to noisy/corrupted images (Macovski, 1996; Parrish et al., 2000; Wiest-Daesslé et al., 2008; Aja-Fernández & Vegas-Sánchez-Ferrero, 2016). Similar to (Upadhyay

| Methods | Metrics | | | | | | | | | |
|---|---|---|---|---|---|---|---|---|---|---|
| | MAE ↓ | MSE ↓ | PSNR ↑ | SSIM ↑ | C.Coeff. ↑ | UCE↓ | R.UCE.↓ | Log-likeli. ↑ | ECE ↓ | Sharp. ↓ |
| $T_2 = 0, T_3 = 0$ | 0.693 | 0.414 | 37.15 | 0.955 | 0.189 | 0.581 | 0.512 | -0.36 | 1.45 | 2.73 |
| $T_2 = 10, T_3 = 10$ | 0.667 | 0.396 | 37.33 | 0.958 | 0.184 | 0.577 | 0.503 | -0.31 | 1.42 | 2.24 |
| $T_2 = 100, T_3 = 0$ | 0.614 | 0.384 | 37.72 | 0.961 | 0.062 | 1.169 | 0.833 | -0.41 | 1.77 | 2.82 |
| $T_2 = 0, T_3 = 100$ | 1.395 | 7.274 | 20.19 | 0.793 | 0.219 | 3.733 | 2.442 | -0.44 | 2.12 | 3.11 |
| $T_2 = 100 \downarrow, T_3 = 0$ | 0.612 | 0.344 | 37.76 | 0.961 | 0.077 | 0.983 | 0.797 | -0.27 | 1.03 | 1.35 |
| $T_2 = 0, T_3 = 100 \downarrow$ | 0.632 | 0.388 | 37.71 | 0.960 | 0.442 | 0.152 | 0.116 | -0.20 | 0.85 | 0.98 |
| $T_2 = 100 \downarrow, T_3 = 100 \downarrow$ | 0.618 | **0.351** | 37.87 | 0.962 | **0.518** | **0.104** | **0.083** | -0.16 | **0.74** | **0.83** |
| $T_2 = 100 \downarrow, T_3 = 100 \downarrow$ with $P(\theta) = \mathcal{N}(\theta)$ | 0.625 | 0.358 | 36.98 | 0.952 | 0.488 | 0.168 | 0.133 | -0.24 | 1.12 | 1.47 |
| $T_2 = 100 \downarrow, T_3 = 100 \downarrow$ with $P(\theta) = \mathcal{E}(\theta)$ | **0.612** | 0.353 | **37.92** | **0.966** | 0.503 | 0.118 | 0.102 | **-0.15** | 0.83 | 1.01 |

Table 2: Ablation study of temperature hyperparameters of the temperature-dependent likelihood used in the proposed *likelihood annealing* (LIKA) method on image super-resolution task.

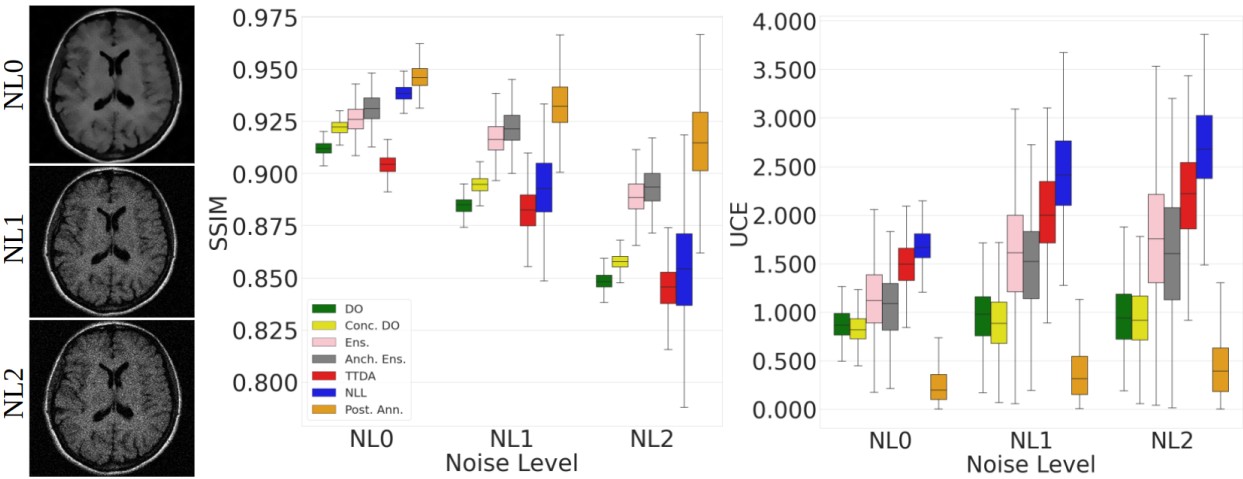

Figure 6: Evaluation of different methods using out-of-distribution input samples for MRI translation.

et al., 2021a;c; Sudarshan et al., 2021), we study the performance of various uncertainty-aware models in the presence of noisy input samples (corrupted with varying degrees of noise) at test time. Figure 6-(left) shows the example of in-distribution (noise-level 0, NL0) and out-of-distribution samples (NL1 and NL2). The severity of corruption gradually increases from NL0 to NL2. From Figure 6-(middle and right), that shows the regression and quality of uncertainty estimates in the presence of OOD samples, we observe that the performance of various models degrades as severity of corruption increases from NL0 to NL2, however our LIKA method performs much better than the compared methods even at higher severity of corruption both in terms of regression and uncertainty calibration metric.

## 5    Conclusion

This paper introduces a novel approach to improve the calibration of uncertainty estimates for regression tasks. We propose a temperature-dependent likelihood that allows for faster and more accurate learning, while avoiding the need for post-hoc calibration. Our method employs a temperature annealing technique during training, which has been shown to lead to 1.5 to 6 times faster convergence compared to existing approaches. Additionally, we demonstrate the effectiveness of our method in producing superior regression results with better calibrated uncertainty estimates, compared to five existing uncertainty estimation methods, across multiple datasets. We further investigate the potential of our approach in out-of-distribution scenarios, showing its ability to generalize well and highlighting its robustness. Our study also includes an ablation analysis, revealing key components of our method and providing valuable insights for future research in uncertainty estimation. Overall, our proposed temperature-dependent likelihood represents a promising direction for improving the efficiency and accuracy of uncertainty estimation in regression tasks.

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
