# OpenReview forum: "Likelihood Annealing: Fast Calibrated Uncertainty for Regression"
_TMLR — Rejected by TMLR_

### Review · Reviewer_pgGX · 2023-03-10

**Summary Of Contributions:**

The authors focus on the problem of improving calibration in regression tasks. For this, they modify the common heteroscedastic Normal likelihood with two additional factors, one giving a further initial focus on the MSE reduction and another encouraging calibration.


**Audience:**

Yes

**Claims And Evidence:**

No

**Requested Changes:**

- A clarification of the problems within Sec 3.1,3.2
- Reformulation of the proposed approach as a modified maximum likelihood optimization scheme instead of the current posterior framing.

**Strengths And Weaknesses:**

## Strengths
The method is compared on a wide range of tasks against a wide range of baselines. However, as discussed in the weaknesses, there seems to be some misunderstanding from my side wrt the proposed method which is why I cannot trust the claimed results so far.

## Weaknesses
This review is to be read primarily as temporary and I hope to converge with the authors to a common understanding as I seem to not understand a couple of fundamental parts of the paper (or vice versa the authors).

- Focusing first on Sec 1 and in particular Fig 1. Can the authors specify what we see here? As I understand it, the blue curve plots the loss of a heteroscedastic negative log-likelihood ignoring constant terms, i.e., $z = u^2/(2v^2) + 0.5 \log(v^2)$. But for a fixed $v$, this is a simple second-order function, i.e., a parabola and not at all the shape given in that Figure. Its gradient is then a linear function scaling with the residuals, i.e., a high residual gives a high gradient as is to be expected. Can the authors explain the discrepancy between the plot and the loss term in (1) as well as the claim that for an MSE regression gradient terms are supposed to be small for large residuals?
See also Fig 3 in this regard. For $T_2, T_3 \to 0$ eq (4) converges to (1), i.e., the proposed orange loss landscape should converge to the blue landscape which it does not.
- For Fig 2 I have a similar problem. What is shown here? In my reading, it should be (3), yet this is a bimodal function, whose optima are not at $|\hat{y} - y| = \hat{\sigma}$. What am I missing?
- Throughout the paper, the authors, e.g., with their title and method name, focus on the term _posterior_ as if they were inferring a posterior that is then marginalized over to arrive at a posterior predictive at. Yet the proposed method and its derivations in Sec 3.2 are _just_ a modified maximum likelihood approach (the authors also acknowledge that they provide maximum likelihood estimation). For most of the paper, they further rely on a _"uniform prior"_, which is improper without ever providing proof that the proposed $p(\theta|D)$ is a proper distribution despite its improper prior. Can the authors comment on their choice of formulation rather than framing it as an MLE approach with modified regularization terms in the log-likelihood objective?
- Until I understand the methodology I cannot properly trust the results although they do indeed look promising. What is lacking throughout the experimental section, however, are any kind of error bars indicating the stability and significance of the differences between the provided results. (Minor question related to that: The Ablation contains experiments with L1/L2 regularization. How are the baselines in the other experiments regularized? Are they unregularized as is the proposed regularization or do they follow some L2 regularization?)

### Minor
- The calibration aspect is formulated throughout the paper as if it were a hard constraint. See e.g., the introduction with _"such that the predicted uncertainty is constrained to be calibrated at every step"_. However, the loss term only contains a weak constraint, not a strict one.
- Relying on a 2d contour plot for visualization instead of the provided 3d plots gives a better overview. Also, the authors should make sure that only valid intervals are shown, e.g., the current plot seems to also show negative standard deviations.
- There are several mathematical errors throughout the paper, e.g., (1) and (4) are wrong. They are only equal up to constant terms.
- Sec 3.1 claims that _"the iid assumption will not capture the heteroscedasticity and will not allow uncertainty estimation"_. The first part of the phrase is correct. But the second part of the sentence is wrong. A homoscedastic noise model still gives us uncertainty in our prediction and it can be estimated just as a heteroscedastic one can. These are just different assumptions about the error in the observations.
- The authors claim to provide an implementation in their supplementary material, which is not provided.
- The reference section points to multiple arxiv preprints that are published papers and should be cited as such.
- Why are the temperatures named $T_2, T_3$ instead of $T_1,T_2$?


## Typos:
- p3: _the optimal parameters **are** learnt_
- p3: _**T**he MLE estimates are_
- Sec 3.3: first line is not properly formatted

---

> ### Author Response · Authors · 2023-04-11
> **Response to reviewer pgGx**
>
> We sincerely thank the reviewer for their effort and for pointing us toward improving the manuscript.
>
> As suggested by the reviewer, we have revamped the Methods section (Section 3.1, 3.2, 3.3) to switch to a likelihood formulation and have updated our visualizations to fix minor errors and enhance clarity. ***We show the updated text in BLUE in the revised manuscript***.
>
> We summarize our revision to various concerns below:
>
> **Understanding Figure 1**
>
> We thank the reviewer for pointing out that for a fixed $v$, the curve for NLL of heteroscedastic Gaussian (i.e., $z = \frac{u}{2v^2} + \frac{\log v^2}{2}$) should look like a parabola -- We realize that we had exchanged labeling of axis $v$ and $u$ in the previous version of the figure. As a result, the parabola was not visible. We updated our figure in the manuscript. We show the 3D plot and the 2D slice, indicating the parabola for improved clarity. Moreover, we want to emphasize that at the beginning of the training phase, both the error and predicted variance are high, which leads to a smaller gradient for the NLL of heteroscedastic Gaussian compared to the temperature-dependent regularizer we introduced, shown more clearly in Figure 1-(Right) (orange vs. blue curve). Therefore, the newly introduced regularizer is helpful.
>
> **Understanding Figure 2**
>
> We thank the reviewer for pointing out the confusion caused by Figure 2, which was meant to be a conceptual schematic (of just the temperature-dependent terms) and not a quantitative figure connected to Equation 3. But, as suggested in the review, it is much clearer to tie it to an equation. We have done the same in the revised manuscript, where we show it for Equation 5 (i.e., $\mathcal{L}_{\text{reg}} = T_2(|\hat{\mathbf{y}}-\mathbf{y}|^2) + T_3 (|\hat{\mathbf{\sigma}} - |\hat{\mathbf{y}} - \mathbf{y}||^2)$, which is the temperature-dependent part of Equation 4) as a function of $\hat{\mathbf{y}}$ for fixed temperatures and variance. This encourages the prediction to be
> close to ground truth and the uncertainty estimate to be close to the error, i.e., calibrated, throughout the training.
>
> **Likelihood vs. Posterior Formulation**
>
> We thank the reviewer for proposing a clearer way to formulate our work (i.e., as a modified likelihood). We concisely explain our previous rationale and then describe how we have accommodated the review comments in the revised manuscript.
>   * We did not emphasize the newly introduced temperature-dependent terms (i.e., the last two terms in Equation 4) as regularizers because they were not purely dependent on the parameters of the network, which is how regularizers are typically defined.  Earlier, we thought the best way to explain this was not to associate this term with prior or likelihood but to introduce it in the posterior as a whole. And the previous version of the manuscript was an attempt to do the same. However, we now realize the shortcomings of this approach.
>
>   * The comments in the review prompted us to explore the alternate formulation. We took inspiration from **[a,b]** to design an *improper likelihood* and use *improper negative log-likelihood* to derive the objective function for training the deep neural network. This formulation does not rely on prior and posterior (but, of course, in the future, can be connected to them). We have revamped Section 3.2 to reflect the same. We have also improved the Figures and their captions so that there is no confusion when referring to the NLL of Gaussian and temperature-dependent regularizers. We believe this improves the overall readability of the manuscript substantially. We thank the reviewer.
>
> **[a]** Pietro and Hennig. "Robust improper maximum likelihood: tuning, computation, and a comparison with other methods for robust Gaussian clustering." Journal of the American Statistical Association, 2016
>
> **[b]** Pietro and Hennig. "Consistency, breakdown robustness, and algorithms for robust improper maximum likelihood clustering." Journal of Machine Learning Research, 2017
>
>
> **Addressing other comments**
>
>  * We clarify in the revised manuscript that our proposed solution is not a hard constraint for calibration. However, it encourages calibration throughout the training.
>  * We have updated our plots to contain only the valid intervals for predicted variance and standard deviation (i.e., only positive). We have also introduced 2D plots where necessary to enhance the clarity. We updated our equations such that they are mathematically precise.
>  * We also point to Figure 6, which shows the error bars for all the methods for the medical imaging experiment; we observe a similar trend for all the experiments.
>  * We have also corrected the sentences pointed out in the review such that they are logically and grammatically correct. And have added the supplementary material.
> * The temperatures are named $T_2, T_3$ because they are the second and third terms (the first term being NLL of Gaussian) in Equation 4.

---

### Review · Reviewer_7FEe · 2023-03-17

**Summary Of Contributions:**

In this work the authors consider the task of regression where learning the variance of the observed variable is of interest. More specifically, the authors consider neural networks mapping features to the mean and variance parameters of a Gaussian. Learning the parameters of the neural network through maximum likelihood results in a challenging optimization landscape, and the authors propose to add two regularizers to the objective: one penalizing prediction errors (much like the Gaussian log-likelihood already does), and another encouraging the average error to match the standard deviation. The authors claim that adding these terms alleviates optimization issues, leading to faster and improved convergence. The weight terms associated with the regularizers are annealed to $0$ throughout training, so that the final objective is still the Gaussian log-likelihood. The authors claim their method is Bayesian (which I mostly disagree with, more details later), and show that it produces good uncertainty estimates.

**Audience:**

Yes

**Broader Impact Concerns:**

I have no broader impact concerns for this paper.

**Claims And Evidence:**

No

**Requested Changes:**

Please address points 3-5 in the weaknesses section of my review. Successfully doing so would involve rewriting the paper to not highlight it as a Bayesian method, compare against [1], and include a better discussion about uncertainty quantification (particularly about aleatoric and epistemic uncertainties); or of course convincing me that I am wrong about these points.

**Strengths And Weaknesses:**

## Strengths ##

1. This paper considers an important problem: parameterizing Gaussians with the output of neural networks is common throughout machine learning, and training instabilities can arise when the variance is learnable.

2. The proposed solution seems to work decently well, and to the best of my knowledge, no similar solution had previously been considered.

## Weaknesses ##

I have three major concerns about the paper that I detail below:

3. The authors claim their method is Bayesian, yet their method just adds regularizers to the log-likelihood! There is no attempt at approximating or sampling from a posterior. The authors claim they are obtaining a MAP estimate, which is why they call the procedure Bayesian. I think this is a stretch, particularly because they use a uniform prior (or if we want to be formal, they consider the limit over uniform priors, as using a uniform prior would impose a bound on the parameter values; although this technicality is missed in the paper), which essentially means just considering the likelihood (it should be pointed out that the regularizers used do not depend only on the parameters, and can thus not be considered as coming from the prior). Furthermore, it is not clear what the actual likelihood is once the regularizers have been added, i.e. in equation 2, it is unclear that the terms after $P(\theta)$ integrate to $1$ when integrated over $\hat{y}_i$. If this object is indeed a valid likelihood, the authors should also further study this distribution if they still want to push the Bayesian view of the method. Overall, I really think this paper should no be understood through a Bayesian lens, but rather as a better way to obtain maximum-likelihood estimates of neural-network-parameterized Gaussians.

4. Following up on the point above, comparisons against methods aiming to alleviate the optimization woes that arise from having neural networks learn Gaussian parameters are completely lacking, and I think those would actually constitute the fairest baselines for comparison, instead of more methods for uncertainty quantification. In particular [1] comes to mind (although the paper is recent, I think it's old enough for a reviewer to reasonably request a comparison; I'll also highlight to the authors that I am not an author of [1]), and for example, comparisons using Gaussian VAEs could also be interesting.

5. This is a paper about uncertainty quantification, yet there is no discussion about the type of uncertainty being quantified. The method that the authors consider quantifies only aleatoric uncertainty, not epistemic uncertainty. This is never explicitly mentioned, and made further confusing by the fact that the authors claim their method is Bayesian. This is not only a conceptual issue I have with the paper: in the experiments, the authors compare against Bayesian methods (more specifically, MC dropout and concrete dropout, whose Bayesian interpretation has been debated before), but I see these methods as being orthogonal: one might very well be Bayesian while adding the regularizers that the authors are using. Similarly, comparing against ensembles is not quite an apples to apples comparison if one expects the ensembles to quantify epistemic uncertainty: one might also train an ensemble of the models that the authors proposed! The same can be said about test-time data augmentations, which is another baseline considered by the authors. In this view, I think the only fair comparison in the paper is the one made against the log-likelihood without any regularizers (which the authors do beat, but once again, I think [1] should be compared against).

Finally, some minor points:

- "architectures including, multilayer perceptrons..." -> "architectures, including multilayer perceptrons ..."
- please mention conformal predictions in the related work section when talking about quantifying uncertainty
- please use \citet{} and \citep{} appropriately in LaTeX
- Left of figure 1: ". the..." -> ". The..."
- "The maximum a posterior" -> "The maximum a posteriori"
- The title in Fig 3 is oddly large
- There's a half line skip at the beginning of section 3.3
- "In Chaotic System using Lorenz Attractor (Lorenz Attractor), the Lorenz equations..." is an odd sentence
- Please explain what T1 and T2 are in the context of MRI. Also, the naming here is unfortunate given the use of $T_2$ and $T_3$ as regularization weights
- It would also be interesting, as per point 4 above, to include the Gaussian log-likelihood achieved by POSTA in Figure 4, rather than the loss which includes regularizers
- The legend is Figure 4 is too small
- Please specify what the final values of $T_2$ and $T_3$ are after annealing, as well as the annealing schedule. It would also be interesting to include an ablation over how these are annealed in Table 2.


[1] Faithful Heteroscedastic Regression with Neural Networks, Stirn et al., 2022

---

> ### Author Response · Authors · 2023-04-11
> **Response to reviewer 7FEe**
>
> We sincerely thank the reviewer for their effort and for pointing us toward improving the manuscript.
>
> As suggested by the reviewer, we have revamped the manuscript (Section 3.1, 3.2, 3.3) to not look at the method from a Bayesian lens but update it accordingly to a modified likelihood formulation. We added the required comparison suggested in the review and show that it can be derived as a *special case ($T_2 = 1, T_3 = 0$ throughout the training)* of our work. We also added a discussion on different types of uncertainty (page 2) and have updated our Figures and Captions to make them more thorough and enhance clarity.
>
> ***We show the updated text in BLUE in the revised manuscript***.
>
> We summarize our revision to various concerns below:
>
> **Replacing the Bayesian lens**
>
> We now realize that explaining our method via *posterior* has shortcomings, as pointed out in the review. Therefore, we have switched to a likelihood formulation where we propose an *improper likelihood* (taking inspiration from **[a,b]**) to derive an objective function for training deep neural networks. Please see Section 3 of our revised manuscript in blue for the details.
>
> **[a]** Pietro and Hennig. "Robust improper maximum likelihood: tuning, computation, and a comparison with other methods for robust Gaussian clustering." Journal of the American Statistical Association, 2016
>
> **[b]** Pietro and Hennig. "Consistency, breakdown robustness, and algorithms for robust improper maximum likelihood clustering." Journal of Machine Learning Research, 2017
>
>
> **Comparing against [1]**
>
> We first observe that in our revised *improper likelihood* is given by,
>
> \\begin{align}
>     P( \mathcal{D} | \theta ) & =  \prod_{i=1}^{i=N}
>      \frac{
>      e^{\frac{-|\hat{\mathbf{y}}_i-\mathbf{y}_i|^2}{(2 \hat{\mathbf{\sigma}}_i^2)}}
>      }{\sqrt{2 \pi \hat{\mathbf{\sigma}}_i^2}}
>     \times
>     e^{-T_2(|\hat{\mathbf{y}}_i-\mathbf{y}_i|^2)}
>     \times
>     e^{-T_3 \left(\begin{array}{l}
>         |\hat{\mathbf{y}}_i - (\mathbf{y}_i+\hat{\mathbf{\sigma}}_i)|^2, \hat{\mathbf{y}}_i \geq \mathbf{y}_i \\\\
>         |\hat{\mathbf{y}}_i - (\mathbf{y}_i-\hat{\mathbf{\sigma}}_i)|^2, \hat{\mathbf{y}}_i < \mathbf{y}_i
>         \end{array}\right)}
> \\end{align}
>
> In the above, if we set $T_2 = 1, T_3 = 0$ and do not anneal them during training, we get the following *improper likelihood*,
>
> \\begin{align}
>     P( \mathcal{D} | \theta ) & =  \prod_{i=1}^{i=N}
>      \frac{
>      e^{\frac{-|\hat{\mathbf{y}}_i-\mathbf{y}_i|^2}{(2 \hat{\mathbf{\sigma}}_i^2)}}
>      }{\sqrt{2 \pi \hat{\mathbf{\sigma}}_i^2}}
>     \times
>     e^{-(|\hat{\mathbf{y}}_i-\mathbf{y}_i|^2)}
> \\end{align}
>
> The NLL of the above results in Equation 5 (objective to train the network) of the suggested comparison [1], give by,
> \begin{align}
>     \sum_{i=1}^{i=N}
>     \frac{\log{\hat{\mathbf{\sigma}}_i^2}}{2} + \frac{|\hat{\mathbf{y}}_i-\mathbf{y}_i|^2}{2 \hat{\mathbf{\sigma}}_i^2}
>     + |\hat{\mathbf{y}}_i-\mathbf{y}_i|^2
> \end{align}
> The above equation, along with the stop gradient operation, is what [1] proposed. Therefore [1] can be captured as a special case of our proposed formulation. Moreover, we show the results for [1] on Boston housing and Atoms3D dataset in Table 1 (denoted as NLL-FH). While the regression performance is comparable to our method, the calibration performance is low (as indicated by higher UCE). This is also expected as the term corresponding to $T_3$ (*missing in [1]*) encourages the variance to be close to error throughout the training, leading to better-calibrated uncertainty estimates.
> We wanted to study the calibration aspect for different methods, so we chose baselines from different families.
>
> **Discussion on Aleatoric and Epistemic Uncertainty**
>
> We have discussed differences in various uncertainty and our focus on aleatoric uncertainty (see blue on page 2/3).
>
> **Addressing other comments**
>
> We fixed the typos and have improved our figures to enhance clarity throughout.

---

> > ### Comment · Reviewer_7FEe · 2023-04-12
> > **Discussion**
> >
> > Thanks to the authors for their reply, I believe the updated manuscript is much improved over the original submission. That being said, I still have some unaddressed concerns:
> >
> > 1. On the use of improper likelihoods: The papers you are referencing about improper likelihoods [a,b] consider a "truly improper" likelihood. What I mean by this is that the likelihood they consider has a "uniform" component, and thus integrates to infinity: it is not proportional to an actual likelihood. The improper likelihood being used in the paper is improper simply because you are choosing to ignore the normalizing constant (I have not done the algebra, but clearly your improper likelihood's normalizing constant could be obtained in closed-form). While ignoring the normalizing constant might not be consequential due to the fact that the temperature parameters are annealed during training, I think this is something that should be explored in the paper: does including the normalizing constant improve performance? In other contexts, including ignored normalizing constants does improve performance [2].
> >
> > 2. On the comparisons to [1]: Thank you for including these, it is promising that LIKA outperforms this baseline. I do have some questions/comments though: (a) Why are comparisons only included for 2 datasets? (b) When talking about [1], you say that "using similar network as (Kendall & Gal, 2017)", what do you mean by this? Are you not using the same network architecture across all your comparisons? (c) The point you bring up about [1] being equivalent to a specific hyperparameter choice of LIKA plus a stop gradient operation is very interesting. It does beg the question though, can LIKA's performance be further improved by also using the stop gradient operation?
> >
> > 3. The concern from point (5) in my original review has not really been addressed: I still don't think all the comparisons are apples-to-apples, and there's no discussion about this. For example, the deep ensembles paper [3] (which by the way I just realized you are citing as a 2016 arxiv preprint but is actually a NeurIPS 2017 paper -- please correct this and go over your references to make sure you are citing the appropriate versions of papers) does not only average multiple mean estimates: each individual member of the ensemble is a Gaussian with a parameterized mean AND variance. It is not clear exactly how you are treating deep ensembles here, but if your ensembles do not output a variance, then the ensemble would be providing an epistemic uncertainty estimate, whereas LIKA's is aleatoric. Again, I don't think these are comparable. And even if you are using ensembles that output variances, these models overall estimate predictive uncertainty, combining both aleatoric and epistemic uncertainty. For a fair comparison against standard deep ensembles, one would have to ensemble LIKA models, to show that the proposed objective also helps when quantifying epistemic uncertainty. The same applies to MC dropout.
> >
> > Finally, the labels in figure 2 are extremely small and hard to read, please update the figure.
> >
> >
> >
> > [2] The continuous Bernoulli: fixing a pervasive error in variational autoencoders, Loaiza-Ganem and Cunningham, 2019
> >
> > [3] Simple and scalable predictive uncertainty estimation using deep ensembles, Lakshminarayanan et al., 2017

---

> > > ### Author Response · Authors · 2023-04-16
> > > **Response to discussion for 7FEe - part 1**
> > >
> > > We thank the reviewer for providing suggestions to improve the manuscript further.
> > >
> > > We address the concerns below, along with updates in the manuscript, in blue, where necessary.
> > >
> > > **On *Improper* vs *Proper* Likelihood**
> > >
> > > As rightly mentioned in the review, in this case, the normalizing constant is inconsequential because of annealing and can be safely ignored. But the review prompted us to try normalization anyways. However, we discover that including a normalization constant to derive an objective function to train DNN is not feasible; we show our work below:
> > >
> > > We want to normalize
> > >
> > > $
> > >      e^{\frac{-|\hat{\mathbf{y}}-\mathbf{y}|^2}{(2 \hat{\mathbf{\sigma}}^2)}}
> > >     \times
> > >     e^{-T_2(|\hat{\mathbf{y}}-\mathbf{y}|^2)}
> > >     \times
> > >     e^{-T_3
> > >     \left(
> > >         \begin{array}{l}
> > >         |\hat{\mathbf{y}} - (\mathbf{y}+\hat{\mathbf{\sigma}})|^2, \hat{\mathbf{y}} \geq \mathbf{y} \\\\
> > >         |\hat{\mathbf{y}} - (\mathbf{y}-\hat{\mathbf{\sigma}})|^2, \hat{\mathbf{y}} < \mathbf{y}
> > >         \end{array}
> > >     \right)
> > >     }
> > > $
> > >
> > > The normalization constant, say $Z$, is given by,
> > >
> > > $
> > > Z = \left( \large{\large{\int_{-\infty}^{\infty}}} \small e^{\frac{-|\hat{\mathbf{y}}-\mathbf{y}|^2}{(2 \hat{\mathbf{\sigma}}^2)}}
> > >     \times
> > >     e^{-T_2(|\hat{\mathbf{y}}-\mathbf{y}|^2)}
> > >     \times
> > >     e^{-T_3
> > >     \left(
> > >         \begin{array}{l}
> > >         |\hat{\mathbf{y}} - (\mathbf{y}+\hat{\mathbf{\sigma}})|^2, \hat{\mathbf{y}} \geq \mathbf{y} \\\\
> > >         |\hat{\mathbf{y}} - (\mathbf{y}-\hat{\mathbf{\sigma}})|^2, \hat{\mathbf{y}} < \mathbf{y}
> > >         \end{array}
> > >     \right)
> > >     } d \hat{\mathbf{y}}
> > >   \right)^{-1}
> > > $
> > >
> > > Let us denote the integral as $I$, i.e.,
> > >
> > > $I = \large{\large{\int_{-\infty}^{\infty}}} \small e^{\frac{-|\hat{\mathbf{y}}-\mathbf{y}|^2}{(2 \hat{\mathbf{\sigma}}^2)}}
> > >     \times
> > >     e^{-T_2(|\hat{\mathbf{y}}-\mathbf{y}|^2)}
> > >     \times
> > >     e^{-T_3
> > >     \left(
> > >         \begin{array}{l}
> > >         |\hat{\mathbf{y}} - (\mathbf{y}+\hat{\mathbf{\sigma}})|^2, \hat{\mathbf{y}} \geq \mathbf{y} \\\\
> > >         |\hat{\mathbf{y}} - (\mathbf{y}-\hat{\mathbf{\sigma}})|^2, \hat{\mathbf{y}} < \mathbf{y}
> > >         \end{array}
> > >     \right)
> > >     } d \hat{\mathbf{y}}$
> > >
> > > We observe that to compute $I$, we can re-write the integration in two ranges ($-\infty, \mathbf{y}$) and ($\mathbf{y}, \infty $), i.e.,
> > >
> > > $I = I_1 + I_2 = \large{\large{\int_{-\infty}^{\mathbf{y}}}} \small e^{\frac{-|\hat{\mathbf{y}}-\mathbf{y}|^2}{(2 \hat{\mathbf{\sigma}}^2)}}
> > >     \times
> > >     e^{-T_2(|\hat{\mathbf{y}}-\mathbf{y}|^2)}
> > >     \times
> > >     e^{-T_3(
> > >         |\hat{\mathbf{\sigma}} - ( \mathbf{y} - \hat{\mathbf{y}})|^2
> > >     )
> > >     } d \hat{\mathbf{y}} +
> > > \large{\large{\int_{\mathbf{y}}^{\infty}}} \small e^{\frac{-|\hat{\mathbf{y}}-\mathbf{y}|^2}{(2 \hat{\mathbf{\sigma}}^2)}}
> > >     \times
> > >     e^{-T_2(|\hat{\mathbf{y}}-\mathbf{y}|^2)} \times
> > >     e^{-T_3( | \hat{\mathbf{\sigma}} - ( \hat{\mathbf{y}} - \mathbf{y} )|^2)}
> > >     d \hat{\mathbf{y}}
> > > $
> > >
> > > We use a symbolic computational tool (*Wolfram Alpha*) to compute each integral.
> > >
> > > We notice that,
> > > $I_1  = \large{\large{\int_{-K \hat{\mathbf{\sigma}}}^{\mathbf{y}}}} \small e^{\frac{-|\hat{\mathbf{y}}-\mathbf{y}|^2}{(2 \hat{\mathbf{\sigma}}^2)}}
> > >     \times
> > >     e^{-T_2(|\hat{\mathbf{y}}-\mathbf{y}|^2)}
> > >     \times
> > >     e^{-T_3(
> > >         |\hat{\mathbf{\sigma}} - ( \mathbf{y} - \hat{\mathbf{y}})|^2
> > >     )
> > >     } d \hat{\mathbf{y}} \text{ with, } K \rightarrow \infty$
> > >
> > > This rewrite allows the library to compute the closed-form solution (the library is unable to compute integral with $\infty$ in limits) given by,
> > >
> > > $I_1 = \frac{\sqrt{\pi }\hat{\mathbf{\sigma}} \exp \left(-\frac{\hat{\mathbf{\sigma}}^2 T_3
> > > \left(2\hat{\mathbf{\sigma}}^2 T_2+1\right)}{2\hat{\mathbf{\sigma}}^2 \left(T_2+T_3\right)+1}\right) \left(\text{erf}\left(\frac{2\hat{\mathbf{\sigma}}^2 T_3}{\sqrt{4\hat{\mathbf{\sigma}}^2 \left(T_2+T_3\right)+2}}\right)-\text{erf}\left(\frac{-2
> > > \hat{\mathbf{\sigma}}^2 T_3 (\mathbf{y}+(K-1)\hat{\mathbf{\sigma}})-\left(2\hat{\mathbf{\sigma}}^2 T_2+1\right) (\mathbf{y}+K\hat{\mathbf{\sigma}})}{\hat{\mathbf{\sigma}} \sqrt{4\hat{\mathbf{\sigma}}^2 \left(T_2+T_3\right)+2}}\right)\right)}{\sqrt{4\hat{\mathbf{\sigma}}^2 \left(T_2+T_3\right)+2}}$
> > >
> > > Taking the limit $K \rightarrow \infty$ and using the fact the $\text{erf}(-\infty)=-1$ and $\text{erf}(\infty)=1$, we get,
> > >
> > > $I_1 = I_2 = \frac{\sqrt{\pi }\hat{\mathbf{\sigma}} \exp \left(-\frac{\hat{\mathbf{\sigma}}^2 T_3
> > > \left(2\hat{\mathbf{\sigma}}^2 T_2+1\right)}{2\hat{\mathbf{\sigma}}^2 \left(T_2+T_3\right)+1}\right) \left(\text{erf}\left(\frac{2\hat{\mathbf{\sigma}}^2 T_3}{\sqrt{4\hat{\mathbf{\sigma}}^2 \left(T_2+T_3\right)+2}}\right)+ 1\right)}{\sqrt{4\hat{\mathbf{\sigma}}^2 \left(T_2+T_3\right)+2}}$
> > >
> > > Therefore, using normalization constant $Z$, in the loss requires the computation of $\text{erf}(\cdot)$ (*again an integral*) at every step, making it infeasible to be used as a normalizing constant.
> > > We will be happy to add this in supplementary.
> > >
> > > ***More responses to be continued as subsequent replies...***

---

> > > > ### Author Response · Authors · 2023-04-16
> > > > **Response to discussion for 7FEe - part 2**
> > > >
> > > > **Addressing other comments**
> > > >
> > > > We highlight that all the compared methods for any given task use the same backbone architecture. Only the head of the network may be split to predict the distribution parameters if it is required for a specific method (e.g., NLL). This is what we intended when we introduced new methods for comparison. We have clarified this further in the manuscript, updated in blue, in the experiments section.
> > > > We also note stop gradient operation did not improve the performance for some preliminary experiments we did (it deteriorated the calibration performance). Finally, we perform the newer experiments on specific datasets due to the time and computational resource constraints.
> > > >
> > > > We have now introduced additional methods from the Dropouts and Ensemble family that predict mean and variances, called DO-NLL and Ens-NLL (details in Blue in the experiments section). We also report the evaluation of aleatoric uncertainty derived from this method and compare it with LIKA in Table 2. We notice that both the regression and uncertainty calibration performance improve over DO/Ens (that captures epistemic uncertainty), but LIKA performance is still superior to DO-NLL/Ens-NLL.
> > > >
> > > > Finally, we fixed the label sizes in Figure 2 and the references.

---

> > > > > ### Comment · Reviewer_7FEe · 2023-04-17
> > > > > **Discussion**
> > > > >
> > > > > Thanks to the authors for their additional replies.
> > > > >
> > > > > About the normalizing constant: I disagree that the normalizing constant involving the erf function implies it is not usable. The erf function is extremely well studied, can be evaluated to machine precision, and its gradients can also be evaluated. Indeed, it is included in all commonly-used deep learning frameworks (PyTorch, JAX, TensorFlow), and it would be rather trivial to include this term in your loss function. Additionally, while I suggested including this term might end up being irrelevant due to the annealing schedule, the same can be said about LIKA to begin with, and believe the authors are being dismissive about this point. In other words, adding a term to the loss, even if annealed to 0 throughout training, might actually change the performance of the final model (or it might not, but this could be easily tested).

---

> > > > > > ### Author Response · Authors · 2023-04-18
> > > > > > **Response to discussion for 7FEe - part 3**
> > > > > >
> > > > > > We thank the reviewer for pointing out the oversight regarding $\text{erf}(\cdot)$ being available readily in the libraries.
> > > > > >
> > > > > > To continue our argument above, the new loss term after including the normalizing constant would be,
> > > > > >
> > > > > > $\mathcal{L} = \sum_{i=1}^{i=N} - \log(Z_i) +  \frac{\log{\hat{\mathbf{\sigma}}_i^2}}{2} + \frac{|\hat{\mathbf{y}}_i-\mathbf{y}_i|^2}{2 \hat{\mathbf{\sigma}}_i^2} + T_2(|\hat{\mathbf{y}}_i-\mathbf{y}_i|^2)  + T_3 (|\hat{\mathbf{\sigma}}_i - |\hat{\mathbf{y}}_i - \mathbf{y}_i||^2)
> > > > > > $
> > > > > >
> > > > > > Which can be simplified to,
> > > > > >
> > > > > > $\mathcal{L} = \sum_{i=1}^{i=N} - \left(\frac{\hat{\mathbf{\sigma}_i}^2 T_3
> > > > > > \left(2\hat{\mathbf{\sigma}_i}^2 T_2+1\right)}{2\hat{\mathbf{\sigma}_i}^2 \left(T_2+T_3\right)+1}\right) + \log \left(\text{erf}\left(\frac{2\hat{\mathbf{\sigma}_i}^2 T_3}{\sqrt{4\hat{\mathbf{\sigma}_i}^2 \left(T_2+T_3\right)+2}}\right)+ 1\right) -  0.5\log \hat{\mathbf{\sigma}}_i^2  + \frac{|\hat{\mathbf{y}}_i-\mathbf{y}_i|^2}{2 \hat{\mathbf{\sigma}}_i^2} + T_2(|\hat{\mathbf{y}}_i-\mathbf{y}_i|^2)  + T_3 (|\hat{\mathbf{\sigma}}_i - |\hat{\mathbf{y}}_i - \mathbf{y}_i||^2)
> > > > > > $
> > > > > >
> > > > > > ***We will present the results with the updated loss function shortly...***

---

> > > > > > > ### Author Response · Authors · 2023-04-18
> > > > > > > **Response to discussion for 7FEe - part 4**
> > > > > > >
> > > > > > > We have now updated the manuscript with an additional section (Section 3.4 in blue) that discusses the proper likelihood, the associated normalizing constant, and the objective derived from the same.
> > > > > > >
> > > > > > > The network trained with the negative log of proper likelihood (i.e., likelihood with normalization constant in our case) is denoted as LIKA-Norm.
> > > > > > > We have updated Table 1 to include LIKA-Norm (in blue) for Boston Housing and Atoms3D dataset. We note that the performance of LIKA-Norm is comparable to LIKA for these two datasets.

---

### Review · Reviewer_FWny · 2023-03-21

**Summary Of Contributions:**

The authors of this paper present a novel method called Posterior Annealing (POSTA) for fast calibrated uncertainty estimation in regression tasks. This method aims to address the slow convergence and poorly calibrated uncertainty estimates often observed in Bayesian deep learning approaches for regression problems. The authors demonstrate the generalizability of their approach across various network architectures, such as multilayer perceptrons, 1D/2D convolutional networks, and graph neural networks. They also evaluate the performance of POSTA on five tasks from different domains. The results show that POSTA enables faster model convergence and provides well-calibrated uncertainty estimates without the need for a post hoc calibration phase.

**Audience:**

Yes

**Broader Impact Concerns:**

Improving methods for calibration estimation is good from a reliability and interpretability standpoint.

**Claims And Evidence:**

Yes

**Requested Changes:**

Requested changes

- It would help to provide more intuition on the two additional terms. Perhaps showing visualizations of 1D slices of the loss would be helpful. Also the $T_2$ term appears to be similar/equivalent to the original term in the minimization, so I'm curious whether we can just scale the first term.
- Tune learning rates for each baseline.
- Run baseline where you vary the temperatures $T_2 = T_3 \neq 100$ to understand how sensitive the algorithm is to choice of temperature.
- I think the figures in figure 4 might not be pdfs since it looks a bit blurry when zoomed in?

Questions
- In figure 3, should there be a plot of $T_2, T_3 = 100, 100$ since those are the default initialization settings?
- Can you explain why the initial convergence is slightly slower in Figure 4 (this contradicts the statement "allows the network to converge faster in the beginning")?


**Strengths And Weaknesses:**

Strengths
- Paper is clear to read and well-motivated.  POSTA enables faster convergence compared to standard Bayesian deep learning approaches, which often suffer from slow convergence in regression tasks.
- POSTA is applicable to various network architectures and benchmarks, improving performance throughout training. The algorithm is easy to implement
- I'm not too well-versed in the literature, but the baselines appears to be comprehensive and capture representative relevant works.

Weaknesses
- "The initial learning rate was set to 2e−4 and cosine annealing was used to decay the learning rate over the course of the learning phase"; the learning rate hyperparameter can have a large effect on the convergence of various algorithms; the results would be stronger if we independently selected a learning rate for each of the baselines (at least in a couple of the evaluations). In general, more discussion on how experimental settings were chosen (that would allow an expert reader to reproduce) would improve the soundness of the results.
- some other suggested changes below

---

> ### Author Response · Authors · 2023-04-11
> **Response to reviewer FWny**
>
> We sincerely thank the reviewer for their effort, for appreciating the problem statement we tackle, and for pointing us toward improving the manuscript.
>
> As suggested by the reviewer, we have revamped Figure 3 to make it more consistent and readable. Moreover, we have also revised other figures to enhance clarity. We have also revised our method section to provide a more intuitive understanding of our loss functions.
>
> ***We show the updated text in BLUE in the revised manuscript***.
>
> We summarize our revision to various concerns below:
>
> * We have revised our methods section to derive our loss function as a modified likelihood instead of trying to associate it with a posterior. This derivation shows a much clearer origin of the objective function. The revised Figure 1 also visualizes the benefits of the proposed objective over the standard objective. To further understand the benefits of the newly added temperature-dependent terms in objective Section 3.2 and Figure 2 provide a thorough discussion on the effect of these newly introduced terms. We also emphasize on the use of a learning rate scheduler (cosine scheduling) that leads to rather robust performance with slight changes in the learning rate, for the problems considered in the paper. This way, using the same configuration for all methods allows proper comparison.
>
> * In Table 2 we have added another experiment for $T_2 = T_3 = 10$. We notice slightly lower performance than our previous configuration, yet better calibration than other baselines.
>
> * We have updated Figure 3 to show annealing starting from $T_2 = T_3 = 100$. We also emphasize that in Figure 4, the proposed method (now called LIKA, orange curve) shows faster convergence than other methods.

---

> > ### Comment · Reviewer_FWny · 2023-04-13
> > **Thanks**
> >
> > Thank you for the detailed updates! One minor change I would suggest is to amend T2 =↓ to T2 =100↓ to make clear the initial temperatures in Table 2.

---

> > > ### Author Response · Authors · 2023-04-16
> > > **Response to discussion for FWny**
> > >
> > > Thank you for your suggestions.
> > >
> > > We have updated the manuscript to accommodate the same.

---

### Decision · Action_Editors · 2023-04-25

**Recommendation:** Reject

**Comment:**

The author discussion period addressed some concerns regarding the conceptual presentation of the method and its interpretation as a probabilistic method.  The last remaining concern is the sharpness of the comparison of uncertainty methods --- ensuring that the kind of uncertainty that is being quantified is being properly compared to other methods that address similar questions.  A more precise set of comparisons would yield more compelling evidence, and shed light on the benefits (and potential drawbacks) of the proposed method. We encourage the authors to make a major submission and resubmit.


**Audience:**

Yes, this work is of interest to the TMLR audience.

**Claims And Evidence:**

Reviewers were appreciative of the author response, yet still had a concern about the strength of the evidence to support their method.  One reviewer explicitly articulated the concern:

> the comparisons are not apples-to-apples: the authors compare their method (LIKA) -- which quantifies aleatoric uncertainty -- against "orthogonal" methods like deep ensembles and MC dropout, which quantify epistemic uncertainty. I think the sensible comparisons here should be LIKA vs NLL, ensemble of LIKA vs ensemble of NLL, etc. The authors have not carried out these comparisons, nor engaged in discussion about this concern with me.